# Enhanced Expert Merging for Mixture-of-Experts in Graph Foundation Models

**Lei Liu**[1,3]**, Xingyu Xia**[1]**, Qianqian Xie**[2,3]**, Ben Liu**[1]**, Wenjie Xu**[1]**, Min Peng**[1,2,3]*

[1]School of Computer Science, Wuhan University, Wuhan, China
[2]School of Artificial Intelligence, Wuhan University, Wuhan, China
[3]Center for Language and Information Research, Wuhan University, Wuhan, China
`{liulei95, summer, xieq, liuben123, vingerxu, pengm}@whu.edu.cn`

## Abstract

Graph foundation models (GFMs) have emerged as a promising paradigm for learning transferable knowledge across diverse graph-structured data. The inherent heterogeneity in features and graph structures poses significant challenges for building scalable and generalizable GFMs. Existing research has employed mixture-of-experts (MoE) models to handle the challenges, assigning the most suitable expert to each graph. Despite this, the underlying mechanisms of MoE within the context of GFMs remain insufficiently explored. In this work, we conduct an in-depth experimental study on an MoE-based GFM and uncover an intriguing finding: the experts ranked second and third assigned by the router perform better than the top-ranked expert. This insight motivates us to investigate the potential of leveraging knowledge embedded across multiple experts. However, directly ensembling the outputs of multiple experts would incur substantial computational overhead, while applying a standard expert merging strategy risks suboptimal performance. To address these challenges, we introduce two enhanced expert merging strategies that retain the computational efficiency of expert merging, while improving performance to approach the effectiveness of expert ensembling. Specifically, we propose (i) a knowledge distillation-inspired expert merging method that aligns the behavior of parameter-fused experts with expert ensembles, and (ii) a theoretical parameter proximity approach that leverages the similarity of expert parameters to approximate ensemble outputs while preserving diversity. Extensive experiments demonstrate that our methods effectively enhance model performance.

## 1 Introduction

Graph-structured data are ubiquitous and appear in diverse domains such as social networks [31], molecular biology [32], recommendation systems [12], and knowledge graphs [9]. As the complexity and volume of such data continue to increase, the development of graph foundation models (GFMs) becomes essential to meet the growing demand for generalizable, scalable, and efficient learning systems capable of handling diverse graph-structured data from various domains. The concept of GFMs mirrors that of foundation models in natural language processing (NLP) and computer vision (CV) [1]: *they are pre-trained on extensive graph data and can be adapted to a wide range of downstream graph tasks* [24]. However, from a technical perspective, GFMs have yet to catch up with their counterparts in language and vision [8].

Inspired by the success of foundation models in NLP and CV, the graph learning community is increasingly focused on developing GFMs [9, 49, 37, 19]. The goal is to learn transferable knowledge

---

*Corresponding author. Email: pengm@whu.edu.cn

39th Conference on Neural Information Processing Systems (NeurIPS 2025).

from broad graph data, enabling their application across various graph domains and downstream tasks. However, building effective GFMs presents two major challenges: (1) **feature heterogeneity**, which arises from the diverse types, semantics, and dimensionalities of node and edge features across graphs, and (2) **structure heterogeneity**, which reflects differences in graph topologies and connectivity patterns, such as variations in local neighborhood density or the presence of scale-free or small-world properties. Currently, there are two main approaches to address feature heterogeneity: (i) Using language models to encode textual attributes from graphs in multiple domains into a unified textual semantic space [23, 15]; (ii) Employing singular value decomposition (SVD) to align the feature dimensions across different graphs [41, 40]. To address structure heterogeneity, recent work has started incorporating mixture-of-experts (MoE) mechanisms into general graph learning models [40].

The effectiveness of MoE has been widely validated in LLMs [5, 17] and multimodal models [22]. However, its working mechanisms in GFMs remain insufficiently explored. To further investigate the role of MoE in GFMs, we conduct exploratory experiments on AnyGraph [40], a recently proposed MoE-based GFM, and discover an interesting phenomenon (shown in Figure 1): the performance of the second and third-ranked experts assigned by the router is stronger than that of the top-ranked expert. This insight motivates us to explore the potential of leveraging knowledge from multiple experts. To this end, we increase the number of selected experts and ensemble their outputs. Experiments reveal that this approach significantly improves the model's performance, which is consistent with findings from previous study [30]. The underlying principle is that consulting multiple experts enables the model to effectively balance and integrate diverse knowledge.

However, directly increasing the number of selected experts significantly increases training and inference costs, particularly for large expert models. A common strategy to mitigate these costs is expert merging, which combines the parameters of the top-$k$ selected experts via weighted averaging [11, 28]. Yet, our findings indicate that this approach often degrades model performance. Such degradation likely stems from the high specialization of learned experts, leading to significant parameter disparities that cause interference during merging [44]. To overcome these challenges, we propose to integrate the efficiency of expert merging with the performance benefits of expert ensembles. Specifically, we introduce two advanced expert merging strategies:

(1) **Knowledge Distillation Enhanced Expert Merging (KDEM):** Since the expert ensemble significantly outperforms the fused expert, we treat the ensemble of the top-$k$ selected experts as the teacher and the parameter-merged expert as the student to align the behavior of the merged expert with that of the ensemble. During training, the collective knowledge of the expert ensemble is distilled into the fused expert, thereby enhancing its capabilities. During inference, the only additional overhead is the weighted averaging of the selected expert parameters, and only the fused expert processes the inputs. This avoids the computational overhead of evaluating multiple experts, significantly reducing inference time. However, ensembling the outputs of multiple experts during training still incurs considerable time costs. Motivated by the observation that real-world teachers do not instruct students continuously, we propose periodically applying knowledge distillation to reduce training time.

(2) **Parameter Proximity Enhanced Expert Merging (PPEM):** To further enhance training efficiency, we undertake a theoretical investigation. Since multi-layer perceptrons (MLPs) constitute the majority of experts in MoE architectures, we demonstrate that when multiple MLPs have similar parameters, their parameter-merged MLP output closely approximates the ensemble of their individual outputs. Based on this theoretical foundation, we propose a strategy to gradually reduce parameter distances between experts. Specifically, we guide the parameters of the top-$k$ experts toward their weighted average, thereby improving mergeability while preserving expert diversity. This method operates directly in parameter space without introducing significant computational overhead.

In summary, our key contributions are as follows:

- To gain deeper insights into the MoE mechanism within graph learning, we examine an MoE-based GFM, revealing the key limitation of the routing strategy: the experts ranked second and third frequently surpass the top-ranked expert. This motivates us to explore expert ensembling, which we find can enhance performance, while a simple expert merging method leads to suboptimal results.

- To effectively and efficiently harness the knowledge of multiple experts, we propose two enhanced expert merging methods: (i) KDEM, which applies knowledge distillation to align the behavior of merged experts with that of expert ensembles. (ii) PPEM, which leverages the theoretical finding to gradually brings the expert parameters closer during training. These methods align the performance of expert merging with that of expert ensembling while preserving computational efficiency.

- We validate our approaches through extensive experiments on 38 graph datasets across various domains and downstream tasks, demonstrating their effectiveness and efficiency.

## 2 Preliminaries

**Graph-Structured Data.** A graph $\mathcal{G}$ can be denoted as $(\mathcal{V}, \mathcal{E})$, where $\mathcal{V} = \{v_i\}$ is the node set and $\mathcal{E} = \{(v_i, v_j)\}$ is the edge set. The topological structure of a graph is often represented in the form of an adjacency matrix $\mathbf{A} \in \mathbb{R}^{|\mathcal{V}| \times |\mathcal{V}|}$, where the elements are 0 or 1, indicating whether an edge exists between the corresponding two nodes. Each node $v_i$ in the graph typically contains a feature vector $\mathbf{x}_i \in \mathbb{R}^{d_0}$, and the feature vectors of all nodes form a feature matrix $\mathbf{X} \in \mathbb{R}^{|\mathcal{V}| \times d_0}$.

**Graph Foundation Models (GFMs).** GFMs aim to learn generalizable graph representations, or graph vocabulary [27, 37], comprising basic transferable units that encode invariant structural and semantic properties across diverse graph datasets. Let $\mathcal{D} = \{\mathcal{G}_i\}_{i=1}^m$ denote a diverse collection of graphs used for training, where each graph $\mathcal{G}_i = (\mathcal{V}_i, \mathcal{E}_i)$ consists of a node set $\mathcal{V}_i$, an edge set $\mathcal{E}_i$, and associated node features $\mathbf{X}_i$, with varying node counts and feature dimensions across graphs. The GFM training process generally involves learning a universal function $f(\mathcal{G}_i; \theta)$, parameterized by $\theta$, that captures key structural and feature-based patterns. Once trained, the GFM can generate embeddings $f(\mathcal{G}_{\text{new}}; \theta)$ for a new graph $\mathcal{G}_{\text{new}}$, enabling broad application to diverse downstream tasks.

**Model Merging and Expert Merging.** Model merging (also called model fusion) refers to the process of combining multiple pre-trained models into a unified one, typically to improve generalization and reduce computational costs in machine learning tasks [20]. One common approach is to merge the parameters of multiple models by performing a weighted average of their learned parameters. Formally, let $\{\mathcal{M}_i(\cdot; \theta_i)\}_{i=1}^k$ represent $k$ distinct pre-trained models, with $\theta_i$ being the parameters of them, then the merged model can be obtained by $\mathcal{M}(\cdot; \bar{\theta}) = \mathcal{M}(\cdot; \sum_{i=1}^k \alpha_i \theta_i)$, where $\alpha_i$ represents the weight assigned to $\mathcal{M}_i$ and $\sum_{i=1}^k \alpha_i = 1$ ensures that the weights are normalized.

Expert merging [11] is a model merging variant applied within the MoE architecture with top-$k$ gating, where the selected top-$k$ expert models are combined or fused. The current expert merging strategy is based on a simple weighted average of the expert parameters. Formally, let $\{\mathcal{M}_k(\cdot; \theta_k)\}_{k=1}^K$ denote $K$ expert models, the merged MoE layer with top-$k$ routing can be expressed as follows:

$$\mathcal{I}, \alpha = g(\mathbf{x}), \quad \mathbf{y} = \mathcal{M}(\mathbf{x}; \bar{\theta}) = \mathcal{M}(\mathbf{x}; \sum_{i \in \mathcal{I}} \alpha_i \theta_i), \tag{1}$$

where $g$ is the gating mechanism (also called the router) which outputs the indices $\mathcal{I} \subseteq \{1, 2, \cdots, K\}$ ($|\mathcal{I}| = k$) and weights $\alpha$ of the selected top-$k$ experts, $\mathcal{M}(\cdot; \bar{\theta})$ is the fused expert model with parameters $\bar{\theta}$, $\mathbf{x}$ and $\mathbf{y}$ represent the input and output of the MoE layer, respectively.

## 3 A Deep Dive into Existing GFM

The rapid advancement of foundation models in NLP and CV has highlighted the need for similar progress in the development of GFMs. Inspired by the success of MoE in language models, researchers have recently begun to apply MoE to GFMs. Nevertheless, the underlying mechanics of MoE in the context of GFMs remain insufficiently explored. To address this gap, we conducted an in-depth experimental investigation of an existing MoE-based GFM to deepen our understanding of MoE in the graph domain. This section begins with a brief introduction to the recently proposed GFM, AnyGraph [40], followed by a detailed description of the experimental explorations and observations.

### 3.1 AnyGraph

AnyGraph is a GFM which uses SVD to address the feature dimension misalignment issue and employs an MoE architecture to handle structure heterogeneity, i.e., assigning a specialized expert to each graph. Specifically, for each graph $\mathcal{G} \in \mathcal{D}$, it first applies SVD to both Laplacian-normalized adjacency matrix $\tilde{\mathbf{A}}$ and feature matrix $\mathbf{X}$, and then combines the dimension-reduced representations:

$$\mathbf{E}_0 = \text{Combine}(\text{SVD}(\tilde{\mathbf{A}}), \text{SVD}(\mathbf{X})) \in \mathbb{R}^{|\mathcal{V}| \times d}, \tag{2}$$

where $d$ is the reduced dimensionality of the features. Subsequently, it injects higher-order connectivity information into $\mathbf{E}_0$ via a non-parametric message passing mechanism [38], producing:

$$\mathbf{E}_1 = \sum_{l=1}^{L} \mathbf{E}_0^{(l)}, \quad \mathbf{E}_0^{(l)} = \tilde{\mathbf{A}} \cdot \mathbf{E}_0^{(l-1)}, \quad \mathbf{E}_0^{(0)} = \mathbf{E}_0, \tag{3}$$

where $L$ is the number of message-passing layers. After this, a non-parametric routing mechanism evaluates all the $K$ experts on $\mathbf{E}_1$, and selects the one with the highest competence score:

$$\hat{\mathbf{E}}_k = \mathcal{M}_k(\mathbf{E}_1; \theta_k), \quad k \in \{1, 2, \cdots, K\}, \tag{4}$$

$$k^\star = \arg\max_k \psi_k, \quad \psi_k = \frac{1}{S} \sum_{s=1}^{S} \text{sigmoid}\left(\hat{\mathbf{e}}_{k,a_s}^\top \hat{\mathbf{e}}_{k,p_s} - \hat{\mathbf{e}}_{k,a_s}^\top \hat{\mathbf{e}}_{k,n_s}\right), \tag{5}$$

where $\mathcal{M}_k$ denotes the $k$-th expert, $k^\star$ is the index of the chosen expert, $\psi_k \in (0, 1)$ denotes the competence score of $\mathcal{M}_k$, $(v_{a_s}, v_{p_s}, v_{n_s})$ are sampled anchor-positive-negative triplets, $S$ is the number of sampled triplets, and $\hat{\mathbf{e}}_{k,i} \in \hat{\mathbf{E}}_k$ is the node embedding of $v_i$ from expert $k$. Finally, the selected expert $\mathcal{M}_{k^\star}$ is applied to obtain the final embedding matrix:

$$\hat{\mathbf{E}} = \mathcal{M}_{k^\star}(\mathbf{E}_1; \theta_{k^\star}), \tag{6}$$

which can be used for downstream tasks. Note that AnyGraph uses the most competent expert, i.e., the top-1 expert. More technical details are provided in Appendix B.1.

## 3.2 Experimental Explorations and Observations

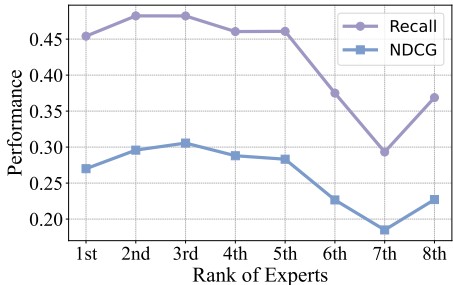

Figure 1: Average test performance of experts ranked by competence score across 18 datasets.

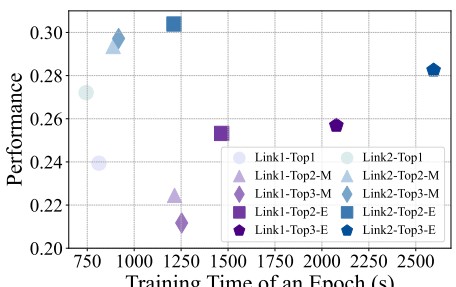

Figure 2: Training time vs. performance of top-1 routing, expert ensemble (-E) and merging (-M).

To gain a deeper understanding of the MoE mechanism in AnyGraph, we conducted a series of empirical studies. Our initial investigation focused on the performance of individual experts, specifically those with varying competence scores $\psi_k$. We directly evaluated the pre-trained expert models—ranked by their competence scores—on the test datasets and observed the following phenomenon:

**Observation 1:** *As illustrated in Figure 1, besides the top-ranked expert, higher-ranked experts also demonstrate substantial performance. For example, the second- and third-ranked experts outperform the top-ranked expert.*

This suggests that the top-$k$ experts collectively possess valuable predictive knowledge. However, AnyGraph exclusively utilizes only the single most competent expert, potentially leaving useful expertise from other high-performing experts untapped. Motivated by this insight, we further explored strategies to incorporate the knowledge embedded in multiple experts. Specifically, we modified the MoE layer to select the top-$k$ experts and aggregate their outputs using weighted averaging (see Appendix B.2 for details). This led to the following observation:

**Observation 2:** *As shown in Figure 2, ensembling the outputs of the top-$k$ experts significantly boosts performance. However, this improvement comes at the cost of increased computational overhead.*

To mitigate this overhead while still leveraging multi-expert knowledge, we turned to recent work on expert merging [11, 28, 51], which suggests that averaging the parameters of the top-$k$ experts can effectively reduce training and inference costs. This is because only a single forward pass through the fused expert is required. Adopting this approach, we made the following observation:

**Observation 3:** *As depicted in Figure 2, while parameter merging reduces time overhead, it may lead to degraded performance in certain cases, such as on the Link1 group. (Link1 and Link2 refer to two dataset groups comprising 15 and 18 datasets, respectively, which will be described in Section 5.1).*

This trade-off raises a key question: *Is it possible to develop a method that combines the efficiency of expert merging with the performance benefits of expert ensembling, thereby enabling both effective and efficient utilization of knowledge from multiple experts?*

## 4  Methods

This section provides an affirmative response to the question. Observation 2 indicates that ensembling the outputs of the top-$k$ experts often yields favorable results. Our key insight is to maintain the efficiency of the parameter-fused expert while aligning its performance more closely with that of the ensemble. To accomplish this, we propose two methods inspired by knowledge distillation and theoretical principles, which are discussed in Sections 4.1 and 4.2, respectively.

### 4.1  Knowledge Distillation Enhanced Expert Merging (KDEM)

Knowledge distillation [14] is a technique that involves transferring knowledge from a large, powerful model (called the teacher model) to a smaller, weaker model (called the student model). The goal is to enable the student model to achieve similar performance to the teacher while being more efficient in terms of computational resources, such as memory and inference time. This aligns perfectly with our goal. Since the performance of the top-$k$ experts' ensemble is excellent, we treat it as a teacher model to teach the parameter-fused expert. This approach allows the knowledge from the top-$k$ experts' ensemble to be transferred to the parameter-fused expert, thereby enhancing its performance. However, traditional knowledge distillation methods have two potential issues: (1)

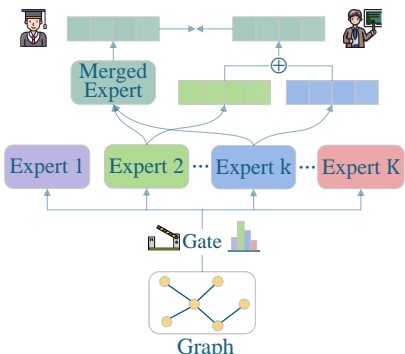

Figure 3: Architecture of KDEM.

The two-stage training paradigm imposes substantial time overhead, since the teacher model must be fully trained before the student can begin training. (2) The top-$k$ experts selected by the router in the pre-trained teacher model may not be consistent with those selected during the training of the student model, leading to potential interference in knowledge transfer.

As demonstrated in Figure 3, we introduce a unified training paradigm that jointly optimizes the teacher and student models to address these issues. Specifically, for a given graph $\mathcal{G}$, after obtaining the initial embeddings $\mathbf{E_1}$ via Eq.(3), the top-$k$ router assigns $k$ experts $\{\mathcal{M}_i(\cdot; \theta_i)\}_{i \in \mathcal{I}}$ to it based on the competence scores $\{\psi_i\}_{i=1}^{K}$, yielding expert indices $\mathcal{I}$ and corresponding weights $\alpha$:

$$\mathcal{I} = \text{Top}k\left(\{\psi_i\}_{i=1}^{K}\right), \quad \alpha = \text{softmax}\left(\{\psi_i\}_{i \in \mathcal{I}}\right). \tag{7}$$

The parameters of these $k$ experts are combined through weighted averaging to form a fused expert. This fused expert generates the outputs of the MoE layer, which are subsequently used to compute a self-supervised loss for training the model:

$$\hat{\mathbf{E}} = \mathcal{M}(\mathbf{E}_1; \bar{\theta}) = \mathcal{M}(\mathbf{E}_1; \sum_{i \in \mathcal{I}} \alpha_i \theta_i), \tag{8}$$

$$\mathcal{L}_{ce} = -\frac{1}{|\mathcal{B}|} \sum_{(v_{a_b}, v_{p_b}, v_{n_b}) \in \mathcal{B}} \log \frac{\exp(\hat{y}_{a_b, p_b} - \hat{y}_{\max})}{\sum_{v_{n_b}} \exp(\hat{y}_{a_b, n_b} - \hat{y}_{\max})}, \tag{9}$$

where $\mathbf{E}_1$ is the embedding matrix from Eq.(3), $\mathcal{B}$ is a mini-batch of triplets, in which $(v_{a_b}, v_{p_b})$ is the positive edge, $(v_{a_b}, v_{n_b})$ is the randomly sampled negative edge, $\hat{y}_{a_b, p_b} = \hat{\mathbf{e}}_{a_b}^{\top} \hat{\mathbf{e}}_{p_b}$ is the inner product, $\hat{\mathbf{e}}_i \in \hat{\mathbf{E}}$ is the embedding of node $v_i$, and $\hat{y}_{\max}$ is the maximum prediction score in the batch, which is used to avoid numerical instability.

To encourage the output of the merged expert to closely resemble the ensemble of the top-$k$ experts while minimizing time overhead, we compute the knowledge distillation loss every few training steps

and incorporate it into the self-supervised loss to form the final loss function:

$$\mathcal{L}_{kd} = \text{MSE}(\hat{\mathbf{E}}, \bar{\mathbf{E}}), \quad \bar{\mathbf{E}} = \sum_{i \in \mathcal{I}} \alpha_i \mathcal{M}_i(\mathbf{E}_1; \theta_i), \tag{10}$$

$$\mathcal{L} = \mathcal{L}_{ce} + \gamma \mathcal{L}_{kd}, \tag{11}$$

where MSE denotes the mean squared error, and $\gamma$ is a tunable parameter that controls the influence of knowledge distillation loss. Notably, we do not explicitly train a separate teacher model. Instead, during training, the ensemble of the top-$k$ experts, $\bar{\mathbf{E}}$, serves as the teacher, distilling its collective knowledge into the fused expert. This distilled knowledge is 'stored' within individual experts through gradient backpropagation and parameter updates. During the forward pass, the knowledge is recovered through expert ensembling. As the expert parameters are updated, the teacher's knowledge is concurrently refined, facilitating subsequent rounds of distillation. Furthermore, the teacher model ($\bar{\mathbf{E}}$) and the student model ($\hat{\mathbf{E}}$) share input samples and gating decisions, ensuring they attend to the same top-$k$ experts for each graph, thereby eliminating any potential inconsistency in expert selection.

## 4.2 Parameter Proximity Enhanced Expert Merging (PPEM)

Although KDEM improves the expressivity of expert merging, it also introduces nontrivial computational overhead. To minimize this cost, we carried out a thorough investigation. Noting that most experts in MoE architectures are instantiated as MLPs [30, 5, 40], we analyzed the theoretical relationship between the output of a parameter-merged MLP and the ensemble of its constituent MLP experts. Our analysis culminates in the following theorem (see Appendix A for a formal proof):

**Theorem 1.** *Given $k$ MLPs $\{f_i(\cdot; \theta_i)\}_{i=1}^k$ with identical architectures and Lipschitz continuous activation function, if their parameters converge pairwise, the merged MLP $f(\cdot; \sum_{i=1}^k \alpha_i \theta_i)$ through parameter average approximates the convex combination of their outputs. Formally, let $\Delta\theta_i = \theta_1 - \theta_i$ measure the parameter deviation from reference parameter $\theta_1$, and $\sum_{j=1}^k \alpha_j = 1$. Then:*

$$\lim_{\|\Delta\theta_i\| \to 0, \forall i \in \{2, \cdots, k\}} \left\| f(\mathbf{x}; \sum_{i=1}^k \alpha_i \theta_i) - \sum_{j=1}^k \alpha_j f_j(\mathbf{x}; \theta_j) \right\| = 0, \tag{12}$$

*where $\|\cdot\|$ denotes the Euclidean norm on the parameter and output spaces.*

Theorem 1 indicates that when the parameters of multiple MLP experts are close to each other, the output of the expert obtained through parameter fusion is also close to the ensemble of those experts. Motivated by this, we aim to make the parameters of the top-$k$ MLP experts selected by the router converge to one another, so that the performance of expert fusion approaches that of expert ensemble. However, if the parameters of all the experts become too close, they will lose their diversity and specialization, which would negatively impact the performance of the MoE architecture.

To address this, we propose a method that brings the top-$k$ experts sufficiently close to leverage the conclusion of Theorem 1, while concurrently maintaining diversity across all experts to preserve the advantages of the MoE architecture. Concretely, we employ an exponential moving average (EMA) to gradually guide the parameters of the top-$k$ experts toward their average:

$$\theta_i \leftarrow \beta\theta_i + (1-\beta)\bar{\theta}, \quad \forall i \in \mathcal{I}, \tag{13}$$

where $\bar{\theta} = \sum_{i \in \mathcal{I}} \alpha_i \theta_i$ and $\beta \in (0, 1)$ controls how rapidly the experts converge: the closer $\beta$ is to 1, the more weight is placed on the expert's previous parameters, resulting in slower updates and more stable behavior. To further regulate this process, we apply the EMA periodically.

This method allows the parameters of the top-$k$ experts selected by the router to gradually move in the same direction, as training progresses, these experts become more similar while retaining their individual specialization, striking a balance between expert similarity and diversity preservation. For instance, over 100 epochs with 6000 iterations each, setting $\beta = 0.999$ and the period of 100, after training, each expert retains only about $0.999^{6000} \approx 0.002$ of its original parameters. By applying the EMA exclusively to the router's current top-$k$ experts while leaving the parameters of the remaining experts unchanged, we preserve the overall diversity and the specialized capabilities of the MoE layer. The training pipeline and time complexity analysis of the two methods are presented in Appendix C.

# 5 Experiments

## 5.1 Experimental Settings

**Datasets** We follow the setup of previous work [40] and use 38 datasets spanning diverse domains. To assess cross-domain generalization ability, we group the datasets in two ways. First, they are split into two major groups, *Link1* and *Link2*, balanced in total and domain-specific edge counts. Second, they are categorized into three domain-specific groups: *E-commerce*, *Academic*, and *Others*. Five datasets are used for the node classification task. Full dataset details are provided in Appendix D.

**Baselines and Evaluation** We compare against eleven baselines spanning four categories; details are provided in Appendix E.1. We adopt a zero-shot setting where models trained on *Link1* group are tested on *Link2* group, and vice versa. Results for each dataset group are averaged based on the number of test samples. For GFM baselines, we use their released pre-trained models; other baselines follow a few-shot setup. Full evaluation procedures are described in Appendix E.2, and implementation details and hyperparameters are documented in Appendix E.3.

## 5.2 Overall Performance

To evaluate the zero-shot prediction capability of our methods, we conduct extensive experiments across various datasets. The evaluation results are shown in Tables 1 and 2. We observe that:

Table 1: Results under zero-shot (AnyGraph, ours) and few-shot (others, 10% train data) settings. For link prediction, we report Recall@20 and NDCG@20; for node classification, we use accuracy and macro-F1 score. The best and second-best results are highlighted in **bold** and underlined, respectively.

| Method | Link1 | | Link2 | | E-commerce | | Academic | | Others | | Node cls. | |
|---|---|---|---|---|---|---|---|---|---|---|---|---|
| | Rec | NDCG | Rec | NDCG | Rec | NDCG | Rec | NDCG | Rec | NDCG | Acc | MacF1 |
| GIN | 11.80 | 5.45 | 21.62 | 13.41 | 13.41 | 8.06 | 20.61 | 9.04 | 18.43 | 11.85 | 36.04 | 30.60 |
| GAT | 13.45 | 6.78 | 15.30 | 8.84 | 9.64 | 5.78 | 11.17 | 4.67 | 16.17 | 20.88 | 54.83 | 41.61 |
| GPF | 6.80 | 3.27 | 16.58 | 9.84 | 18.72 | 10.94 | 14.83 | 6.41 | 4.51 | 3.44 | 16.29 | 16.00 |
| GraphPrompt | 5.42 | 3.11 | 6.10 | 3.62 | 6.06 | 3.36 | 7.72 | 3.40 | 3.42 | 2.72 | 23.15 | 22.89 |
| GraphCL | 20.55 | 10.76 | 31.42 | 19.91 | 26.05 | 14.59 | 28.69 | 14.31 | 24.62 | 15.90 | 48.75 | 36.15 |
| AnyGraph | 23.94 | 12.68 | 46.42 | 27.21 | 26.92 | 15.05 | **32.74** | 15.31 | 46.83 | 28.97 | **64.31** | 43.24 |
| KDEM | 24.11 | 12.80 | **51.69** | **32.60** | **35.05** | **22.02** | 32.15 | 15.21 | **46.86** | **30.00** | 63.56 | **45.03** |
| PPEM | **24.33** | **12.93** | 50.77 | 31.93 | 33.99 | 21.17 | 32.47 | **15.47** | 46.52 | 29.73 | 62.65 | 44.16 |

Table 2: Comparison with other existing GFMs in zero-shot scenario. GR denotes the Goodreads dataset. Since UniGraph uses Arxiv for instruction fine-tuning in the zero-shot setting, we report its few-shot result instead. The GFT result is also under a few-shot setting. As GOFA does not support ranking-based link prediction metrics, we report accuracy to align with its evaluation protocol. The GOFA result on PubMed-link is under a supervised learning setup and is obtained from its paper.

| Method | GraphGPT | | | | OpenGraph | | UniGraph | GFT | GOFA | |
|---|---|---|---|---|---|---|---|---|---|---|
| Data | Cora | | PubMed | | Ecom. w/o GR | | Arxiv | | Cora-link | PubMed-link |
| Metric | Acc | MacF1 | Acc | MacF1 | Recall | NDCG | Acc | | Acc | Acc |
| Baseline | 18.13 | 12.72 | 70.11 | 64.91 | 14.44 | 10.99 | 31.35 | 36.29 | 86.31 | 93.97 |
| KDEM | **62.45** | **56.32** | **70.36** | **68.83** | **28.30** | **20.02** | **61.86** | | 93.03 | 93.30 |
| PPEM | 62.10 | 55.93 | 69.94 | 67.73 | 22.76 | 19.76 | 61.20 | | **93.14** | **94.10** |

(1) MoE-based methods substantially outperform other models. Our methods, alongside AnyGraph, employ the MoE mechanism to assign the most suitable experts to each graph, achieving higher zero-shot capabilities. The MoE's effective handling of heterogeneity across different graphs demonstrates its potential to become a powerful backbone for GFMs. Other methods, such as classical GNNs, require supervised training on specific datasets and exhibit weak generalization capabilities. Pretraining then fine-tuning or prompting paradigms also fail to achieve good performance due to substantial distribution disparities among graph datasets. These methods struggle to learn transferable knowledge under distribution shifts and can even lead to negative transfer issues [3].

(2) Compared to AnyGraph, our proposed methods achieve superior performance. For example, KDEM and PPEM achieve a 5.27% and 4.35% improvement in recall, respectively, on the *Link2* dataset group. The superior performance stems from our model's ability to effectively combine the knowledge of multiple experts, whereas AnyGraph can only utilize a single expert. Incorporating multiple experts offers the following advantages: (i) More accurate and robust predictions by combining diverse expertise; (ii) Reduced risk of overfitting on a single, overly specialized expert. Moreover, our methods yield significantly larger improvements on *E-commerce* datasets compared to those from *Academic* or *Others* domains, suggesting that the *E-commerce* domain benefits more substantially from the integration of multiple experts. The fine-grained experimental results and analyses for each dataset are provided in Appendix F.

## 5.3 Ablation Study

This section demonstrates the effectiveness of the submodules in our model by comparing with ablated variants. As a reference, we present the results of expert ensembling. From Table 3, we can observe that:

(1) Knowledge distillation significantly enhances the performance of expert fusion. The -kd variant, which directly performs a weighted average of the parameters of the selected top-$k$ experts without any additional enhancements, shows degraded per-

Table 3: Effectiveness of the key components.

| Method | Link1 | | Link2 | |
|---|---|---|---|---|
| | Recall | NDCG | Recall | NDCG |
| Expert Ensemble | 25.69 | 13.50 | 48.89 | 30.39 |
| KDEM-kd | 22.42 | 11.66 | 48.34 | 29.72 |
| KDEM | 24.11 | 12.80 | 51.69 | 32.60 |
| PPEM-EMA | 22.42 | 11.66 | 48.36 | 29.62 |
| PPEM | 24.33 | 12.93 | 50.77 | 31.93 |

formance compared to KDEM. This may be attributed to the simple fusion method causing interference [44] among the expert parameters, leading to the loss of some knowledge during expert fusion. In contrast, KDEM improves performance by distilling knowledge from the expert ensembling into the fused expert, guiding it to retain the complementary strengths of each specialist expert.

(2) Encouraging parameter proximity among the top-$k$ experts substantially improves the performance of expert fusion. Grounded in theoretical derivation, PPEM gradually brings the parameters of the selected experts closer together, aligning the output of the fused experts with that of expert ensembling, thereby boosting the overall performance.

(3) Expert merging is a promising method for leveraging the knowledge of multiple experts. Since ensembling multiple experts incurs a significant training overhead, we evaluated only the top-2 and top-3 expert ensembles, and the results shown are the best among these. As demonstrated in the results, KDEM and PPEM achieve performance comparable to expert ensembling on the *Link1* group and even surpass it on *Link2*, highlighting the potential of expert fusion strategies.

## 5.4 Computational Efficiency Analysis

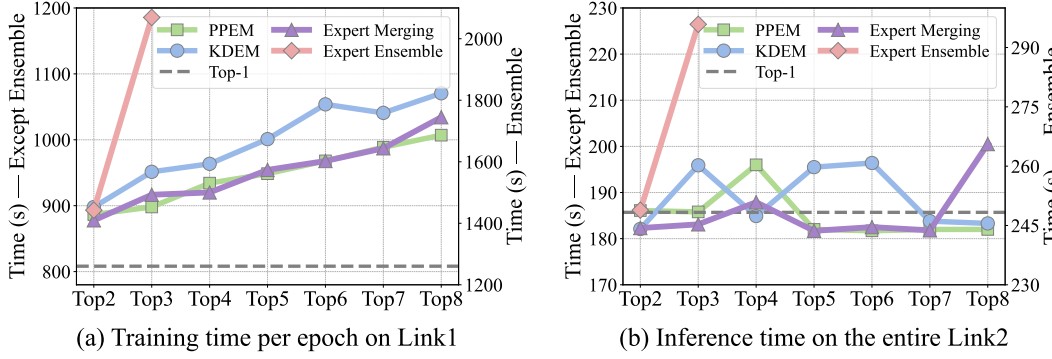

(a) Training time per epoch on Link1          (b) Inference time on the entire Link2

Figure 4: Computational efficiency compared with top-1 routing of AnyGraph, standard expert merging, and expert ensemble (expert ensemble: right y-axis; all other methods: left y-axis).

This section evaluates the computational efficiency of KDEM and PPEM compared to top-1 routing, standard expert merging, and expert ensemble. As depicted in Figure 4, we observe: (1) Relative to

the other methods, the expert ensemble imposes substantial training and inference time overhead. (2) Compared to top-1 routing, using top-$k$ routing inherently increases training overhead, with training time gradually growing as $k$ increases. (3) KDEM incurs a marginal increase in training time over standard expert merging (averaging 4.8%), whereas PPEM exhibits almost no increase. Note that a relatively small $k$ value often suffices for excellent performance (refer to Appendix G.2), so the additional training overhead is limited. (4) The inference times for expert merging methods exhibit only minor fluctuations around that of top-1 routing. These fluctuations are likely due to device measurement variability, indicating that these expert merging methods introduce negligible increases in test time. These findings align with our time complexity analysis in Appendix C.2.

## 5.5 Further Discussion

### 5.5.1 Discussion of Two-Stage Knowledge Distillation and KDEM

As mentioned in Section 4.1, pretraining a teacher model before performing knowledge distillation not only incurs substantial training overhead, but also risks inconsistency between the experts selected during teacher training and those used in the distillation phase, due to the dynamic nature of expert assignment during training. In contrast, KDEM ensures that the teacher and student attend to the same set of experts on the same input, thereby maintaining consistency in the information flow. Moreover, this distillation strategy makes the knowledge transfer process more stable: the teacher's outputs are continuously updated throughout training, allowing the student model to optimize along a convergence path aligned with the teacher. As shown in Table 4, the performance of the two-stage approach—first training the teacher and then distilling—is clearly inferior to our proposed KDEM.

Table 4: Performance comparison between KDEM and the two-stage approach of training a teacher model followed by knowledge distillation.

| Method | Link1 | | Link2 | |
|---|---|---|---|---|
| | Recall | NDCG | Recall | NDCG |
| Two-stage KD | 23.19 | 12.03 | 47.13 | 28.61 |
| KDEM | **24.11** | **12.80** | **51.69** | **32.60** |

### 5.5.2 A Deeper Analysis of PPEM Effectiveness

We posit that expert merging only yields benefits when the top-$k$ experts are both sufficiently similar—so as to minimize parameter interference—and yet retain enough diversity to maximize overall knowledge capacity. To test this, we performed two complementary experiments:

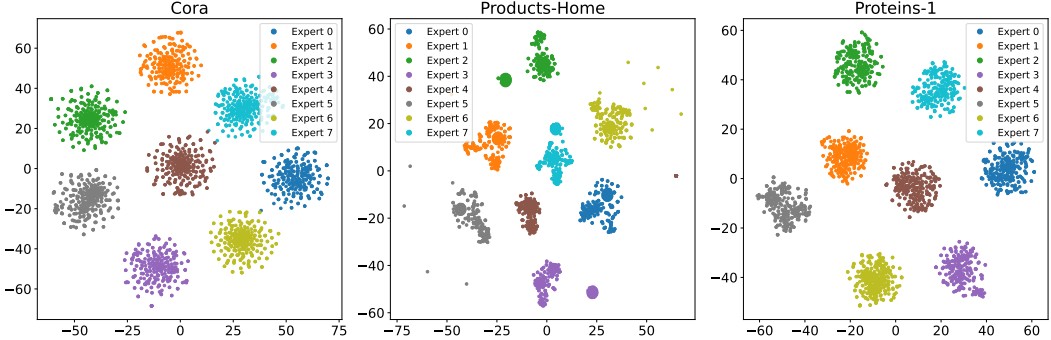

Figure 5: T-SNE visualization of node embeddings generated by all 8 experts trained with PPEM on the Cora (academic), Products-Home (e-commerce), and Proteins-1 (biological) datasets. A random sample of 2,500 nodes is selected from each graph for visualization.

**(1) Expert Similarity Analysis:** We trained two models, one using PPEM and one using the vanilla expert merging strategy, and then measured the cosine similarity between the outputs of the top-2 selected experts on each dataset. Under PPEM, the mean similarity across all test sets was

0.953, compared to just 0.828 for the vanilla approach. The higher similarity under PPEM reduces interference between expert parameters, facilitating a more effective merge of the top-$k$ experts.

**(2) Expert Diversity Verification:** Figure 5 presents a t-SNE visualization of the output representations from all 8 expert models trained with PPEM. Despite the high similarity between the top-2 experts, the experts remain well separated in the embedding space, confirming that diversity is preserved. This maintained heterogeneity allows the merged experts to cover a broader range of knowledge, thereby improving the model's generalization performance.

### 5.5.3 Visualization of Routing Mechanism

We visualize expert routing to intuitively demonstrate its working mechanism. Figure 6 depicts the competence scores of a PPEM model trained on *Link2* and tested on *Link1*. From this visualization, we observe: **(1) Shared expert competence.** Many datasets receive similarly high scores from multiple top-ranked experts. For example, experts 0 and 3 achieve nearly identical scores on the Prod-tech dataset, suggesting that combining knowledge from several experts is advantageous. **(2) Domain-consistent assignments.** Related datasets are routed to the same experts. Notably, both citation datasets assign their highest scores to experts 1, 0, and 6, indicating that the router captures underlying domain similarities. A further discussion of the routing mechanism can be found in Appendix H.

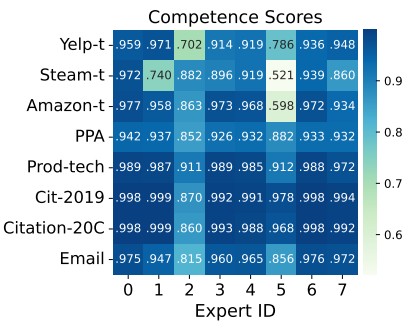

Figure 6: Competence scores of different experts in a top-2 merged PPEM model.

## 6 Related Work

**Graph Foundation Models** GFMs are still in their infancy: most are tailored to specific tasks or domains, with only a handful demonstrating preliminary cross-domain and cross-task transferability. For instance, ULTRA [9] and GraphAny [49] excel at knowledge graph completion and node classification, respectively, while DiG [50] and MiniMol [18] focus on the molecular domain. More versatile methods leverage large language models: OFA [23] generates textual node and edge descriptions and embeds them in a unified space; GraphGPT [33] and LLaGA [4] align graph and text embeddings via projection layers before prediction; UniGraph [13] and GOFA [19] design self-supervised pretraining objectives to learn a graph-language encoder that jointly captures structural and textual information. However, these methods all rely on text features and cannot directly exploit the original graph attributes. AnyGraph [40] addresses structural heterogeneity by dispatching a specialized expert per graph via an MoE layer, enabling improved generalization, but it still underutilizes the complementary knowledge encoded in its multiple experts.

Additional literature relevant to this study is discussed in Appendix J.

## 7 Conclusion

This paper presents an in-depth investigation of an existing MoE-based GFM, revealing an interesting finding: the performance of the second and third-ranked experts assigned by the router exceeds that of the top-ranked expert. This inspires us to explore the use of multiple experts in MoE architectures. Additionally, we find that integrating the outputs of multiple experts incurs significant computational overhead, and directly applying expert merging leads to suboptimal performance. To address these challenges, we propose two enhanced expert merging strategies: (1) using knowledge distillation to align the performance of the merged expert with that of the expert ensemble; (2) based on theoretical principles, using parameter proximity to bring the selected expert parameters closer together. Extensive experiments demonstrate that our methods achieve excellent performance without introducing significant time overhead. This work provides new insights into the role of MoE mechanisms in GFMs and establishes a foundation for efficient expert utilization within MoE architectures. We envision applying our methods to MoE-based large language models, alleviating their computational and inference-time overhead while preserving strong performance.

## Acknowledgments and Disclosure of Funding

This work was supported by Key Project of the National Natural Science Foundation of China (U23A20316) and CCF-Tencent Rhino-Bird Open Research Fund (CCF-Tencent RAGR20250115).

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

# A  Proof of Theorem 1

To prove Theorem 1, we require the following lemma.

**Lemma 2.** *Given the same conditions as in Theorem 1, if the parameters of $k$ MLPs are sufficiently close, the output of the merged MLP through parameter averaging will converge to the output of each individual MLP. Specifically, for any $j \in \{1, 2, \cdots, k\}$,*

$$\lim_{\|\Delta\theta_i\| \to 0, \forall i \in \{2, \cdots, k\}} \left\| f(\mathbf{x}; \sum_{i=1}^{k} \alpha_i \theta_i) - f_j(\mathbf{x}; \theta_j) \right\| = 0. \tag{14}$$

*Proof of Lemma 2.* Without loss of generality, we prove the case for $k = 2$ and $j = 1$; the general case follows similarly.

Let $\theta_1 = \left( \mathbf{W}_1^{(1)}, \mathbf{b}_1^{(1)}, \cdots, \mathbf{W}_1^{(L')}, \mathbf{b}_1^{(L')} \right)$ and $\theta_2 = \left( \mathbf{W}_2^{(1)}, \mathbf{b}_2^{(1)}, \cdots, \mathbf{W}_2^{(L')}, \mathbf{b}_2^{(L')} \right)$ denote the parameters of two MLPs with $L'$ layers, where $\mathbf{W}^{(l)}, \mathbf{b}^{(l)}$ are the weight matrix and bias of the $l$-th layer, respectively. We proceed by induction on the layer index $l$.

**Base Case** ($l = 1$)**:** Let $\tilde{\mathbf{z}}^{(1)}$ and $\mathbf{h}_1^{(1)}$ be the outputs of the merged MLP $f$ and the first MLP $f_1$ at layer 1, respectively. Then:

$$
\begin{aligned}
\left\| \tilde{\mathbf{z}}^{(1)} - \mathbf{h}_1^{(1)} \right\| &= \left\| \sigma \left( \left( \alpha_1 \mathbf{W}_1^{(1)} + \alpha_2 \mathbf{W}_2^{(1)} \right) \mathbf{x} + \alpha_1 \mathbf{b}_1^{(1)} + \alpha_2 \mathbf{b}_2^{(1)} \right) - \sigma \left( \mathbf{W}_1^{(1)} \mathbf{x} + \mathbf{b}_1^{(1)} \right) \right\| \\
&\leq C \left\| \left( \alpha_1 \mathbf{W}_1^{(1)} + \alpha_2 \mathbf{W}_2^{(1)} \right) \mathbf{x} + \alpha_1 \mathbf{b}_1^{(1)} + \alpha_2 \mathbf{b}_2^{(1)} - \left( \mathbf{W}_1^{(1)} \mathbf{x} + \mathbf{b}_1^{(1)} \right) \right\| \\
&= C\alpha_2 \left\| (\mathbf{W}_2^{(1)} - \mathbf{W}_1^{(1)})\mathbf{x} + (\mathbf{b}_2^{(1)} - \mathbf{b}_1^{(1)}) \right\| \\
&\leq C\alpha_2 \left( \left\| \mathbf{W}_2^{(1)} - \mathbf{W}_1^{(1)} \right\| \cdot \|\mathbf{x}\| + \left\| \mathbf{b}_2^{(1)} - \mathbf{b}_1^{(1)} \right\| \right) \\
&\to 0 \quad \text{as } \|\Delta\theta_2\| = \|\theta_1 - \theta_2\| \to 0,
\end{aligned}
\tag{15}
$$

where $\sigma(\cdot)$ is the activation function with Lipschitz constant $C$, and the first inequality arises from the Lipschitz continuity of the activation function.

**Inductive Step** ($l \geq 2$)**:** Assume the claim holds for layer $l - 1$ (i.e., $\|\tilde{\mathbf{z}}^{(l-1)} - \mathbf{h}_1^{(l-1)}\| \to 0$). For layer $l$, the outputs of $f$ and $f_1$ are denoted as follows:

$$\tilde{\mathbf{z}}^{(l)} = \sigma \left( \left( \alpha_1 \mathbf{W}_1^{(l)} + \alpha_2 \mathbf{W}_2^{(l)} \right) \tilde{\mathbf{z}}^{(l-1)} + \alpha_1 \mathbf{b}_1^{(l)} + \alpha_2 \mathbf{b}_2^{(l)} \right), \tag{16}$$

$$\mathbf{h}_1^{(l)} = \sigma \left( \mathbf{W}_1^{(l)} \mathbf{h}_1^{(l-1)} + \mathbf{b}_1^{(l)} \right), \tag{17}$$

then we have

$$
\begin{aligned}
\left\| \tilde{\mathbf{z}}^{(l)} - \mathbf{h}_1^{(l)} \right\| &= \left\| \sigma \left( \left( \alpha_1 \mathbf{W}_1^{(l)} + \alpha_2 \mathbf{W}_2^{(l)} \right) \tilde{\mathbf{z}}^{(l-1)} + \alpha_1 \mathbf{b}_1^{(l)} + \alpha_2 \mathbf{b}_2^{(l)} \right) - \sigma \left( \mathbf{W}_1^{(l)} \mathbf{h}_1^{(l-1)} + \mathbf{b}_1^{(l)} \right) \right\| \\
&\leq C \left\| \left( \alpha_1 \mathbf{W}_1^{(l)} + \alpha_2 \mathbf{W}_2^{(l)} \right) \tilde{\mathbf{z}}^{(l-1)} + \alpha_1 \mathbf{b}_1^{(l)} + \alpha_2 \mathbf{b}_2^{(l)} - \left( \mathbf{W}_1^{(l)} \mathbf{h}_1^{(l-1)} + \mathbf{b}_1^{(l)} \right) \right\| \\
&= C \left\| \mathbf{W}_1^{(l)} \left( \tilde{\mathbf{z}}^{(l-1)} - \mathbf{h}_1^{(l-1)} \right) + \alpha_2 \left( \mathbf{W}_2^{(l)} - \mathbf{W}_1^{(l)} \right) \tilde{\mathbf{z}}^{(l-1)} + \alpha_2 \left( \mathbf{b}_2^{(l)} - \mathbf{b}_1^{(l)} \right) \right\| \\
&\leq C \left( \|\mathbf{W}_1^{(l)}\| \cdot \|\tilde{\mathbf{z}}^{(l-1)} - \mathbf{h}_1^{(l-1)}\| + \alpha_2 \|\mathbf{W}_2^{(l)} - \mathbf{W}_1^{(l)}\| \cdot \|\tilde{\mathbf{z}}^{(l-1)}\| + \alpha_2 \|\mathbf{b}_2^{(l)} - \mathbf{b}_1^{(l)}\| \right) \\
&\to 0 \quad \text{as } \|\Delta\theta_2\| \to 0,
\end{aligned}
\tag{18}
$$

where the last step follows from the inductive hypothesis and $\|\tilde{\mathbf{z}}^{(l-1)}\|$ being bounded (since $\mathbf{x}$ and activation functions are bounded). By induction, the lemma holds for all layers. $\square$

*Proof of Theorem 1.* The expression inside the norm of Eq.(12) can be rewritten as:

$$\sum_{j=1}^{k} \alpha_j \left[ f(\mathbf{x}; \sum_{i=1}^{k} \alpha_i \theta_i) - f_j(\mathbf{x}; \theta_j) \right]. \tag{19}$$

Therefore, Theorem 1 can be proven as follows:

$$\lim_{\|\Delta\theta_i\|\to 0, \forall i\in\{2,\cdots,k\}} \left\| f(\mathbf{x}; \sum_{i=1}^{k} \alpha_i\theta_i) - \sum_{j=1}^{k} \alpha_j f_j(\mathbf{x};\theta_j) \right\|$$

$$= \lim_{\|\Delta\theta_i\|\to 0, \forall i\in\{2,\cdots,k\}} \left\| \sum_{j=1}^{k} \alpha_j \left[ f(\mathbf{x}; \sum_{i=1}^{k} \alpha_i\theta_i) - f_j(\mathbf{x};\theta_j) \right] \right\| \quad \text{(by Eq.(19))}$$

$$\leq \lim_{\|\Delta\theta_i\|\to 0, \forall i\in\{2,\cdots,k\}} \sum_{j=1}^{k} \alpha_j \left\| f(\mathbf{x}; \sum_{i=1}^{k} \alpha_i\theta_i) - f_j(\mathbf{x};\theta_j) \right\| \quad \text{(triangle inequality)} \qquad (20)$$

$$= \sum_{j=1}^{k} \alpha_j \lim_{\|\Delta\theta_i\|\to 0, \forall i\in\{2,\cdots,k\}} \left\| f(\mathbf{x}; \sum_{i=1}^{k} \alpha_i\theta_i) - f_j(\mathbf{x};\theta_j) \right\| \quad \text{(linearity of limits)}$$

$$= 0. \quad \text{(by Lemma 2)}$$

$\square$

**Remark:** Most common activation functions used in neural networks are Lipschitz continuous, with varying Lipschitz constants: ReLU (Lipschitz constant 1), Leaky ReLU (Lipschitz constant $\max(1,\alpha)$ where $\alpha$ is the slope parameter), ELU (Lipschitz constant $\max(1,\alpha)$), Sigmoid (Lipschitz constant 0.25), Tanh (Lipschitz constant 1), Softplus (Lipschitz continuous with constant 1), GELU (Lipschitz continuous but no simple closed-form constant). Thus, our theorem is broadly applicable.

## B  Details of AnyGraph and Observations

### B.1  AnyGraph

For each graph $\mathcal{G} \in \mathcal{D}$, AnyGraph begins by applying SVD to both its Laplacian-normalized adjacency matrix $\tilde{\mathbf{A}} \in \mathbb{R}^{|\mathcal{V}|\times|\mathcal{V}|}$ and its node feature matrix $\mathbf{X} \in \mathbb{R}^{|\mathcal{V}|\times d_0}$, yielding

$$\mathbf{U}_1, \mathbf{\Lambda}_1, \mathbf{V}_1 = \text{SVD}(\tilde{\mathbf{A}}), \quad \mathbf{U}_2, \mathbf{\Lambda}_2, \mathbf{V}_2 = \text{SVD}(\mathbf{X}). \qquad (21)$$

Here, $\mathbf{U}_1, \mathbf{V}_1 \in \mathbb{R}^{|\mathcal{V}|\times d}$ and $\mathbf{U}_2 \in \mathbb{R}^{|\mathcal{V}|\times d}$, $\mathbf{V}_2 \in \mathbb{R}^{d_0\times d}$. The initial node embeddings are then formed by linearly combining these factors and normalizing:

$$\mathbf{E}_0 = \text{LayerNorm}\left( \mathbf{U}_1\sqrt{\mathbf{\Lambda}_1} + \mathbf{V}_1\sqrt{\mathbf{\Lambda}_1} + \text{Flip}(\mathbf{U}_2\sqrt{\mathbf{\Lambda}_2}) \right), \qquad (22)$$

where $\text{Flip}(\cdot)$ denotes reversing the order of the $d$ dimensions in each row. Next, it inject multi-hop connectivity information into $\mathbf{E}_0$ via a simplified GCN module to obtain $\mathbf{E}_1$ (Eq.(3)). The enriched embeddings $\mathbf{E}_1$ are then passed through an MoE layer, where each graph is dispatched to its most competent MLP expert according to the routing rule in Eq.(5).

To alleviate the expert training imbalance issue, AnyGraph employs a *training frequency regularization* method to adjust the competence score as follows:

$$\psi'_k = \psi_k \cdot \left[ 1 + \left( \frac{1}{2} - \frac{m_k}{\sum_{i=1}^{K} m_i} \right) \rho \right], \qquad (23)$$

where $m_k$ is the current training steps of $k$-th expert, $\rho$ is a hyperparameter controlling the strength of recalibration, and $\psi'_k$ is the resulting adjusted score for the $k$-th expert. If an expert is trained too frequently, its competence score is multiplied by a penalty factor.

Finally, the entire model is optimized end-to-end using a self-supervised loss:

$$\mathcal{L}_{ce} = -\frac{1}{|\mathcal{B}|} \sum_{(v_{a_b}, v_{p_b}, v_{n_b})\in\mathcal{B}} \log \frac{\exp(\hat{y}_{a_b,p_b} - \hat{y}_{\max})}{\sum_{v_{n_b}} \exp(\hat{y}_{a_b,n_b} - \hat{y}_{\max})}, \qquad (24)$$

where $(v_{a_b}, v_{p_b}, v_{n_b})$ is an anchor-positive-negative triplet, the term $\hat{y}_{i,j} = \hat{\mathbf{e}}_i^\top \hat{\mathbf{e}}_j$ is the inner product of node embeddings, each $\hat{\mathbf{e}} \in \hat{\mathbf{E}}$ is the output of the chosen expert (Eq.(6)), and $\hat{y}_{\max} = \max_{i,j} \hat{y}_{i,j}$ is used for numerical stability.

## B.2 Experimental Details of Observations

This section provides a detailed explanation of the experiments presented in Section 3.2.

**Observation 1:** We evaluate the performance of experts with different rankings on the test set using the publicly available code and pretrained models released by AnyGraph.[2] The authors provide two pretrained models, trained separately on the *Link1* and *Link2* dataset groups. To assess zero-shot generalization, we test the model trained on one dataset group against the datasets in the other group. Figure 1 presents the results of the model trained on *Link1* and evaluated on the *Link2* dataset group. The results for the reverse setting—trained on *Link2* and testing on *Link1*—are shown in Figure 7. As we can see, although the second- and third-ranked experts do not perform as dramatically as in Figure 1, they remain competitively strong. Moreover, we observe a sudden performance drop as expert rank increases—for instance, from the fourth-ranked to the fifth-ranked expert in Figure 7. The same trend is apparent in Figure 1. This suggests that the top-$k$ experts may all contribute positively to the prediction, while the lowest-ranked few experts may be harmful.

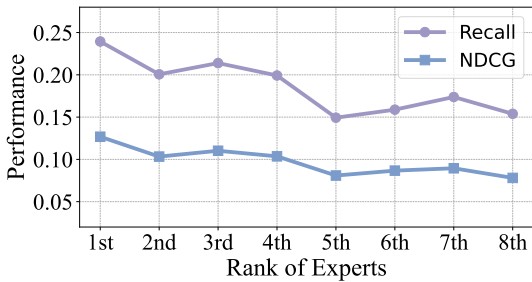

Figure 7: Average test performance of experts ranked by competence score across 15 datasets (*Link1*).

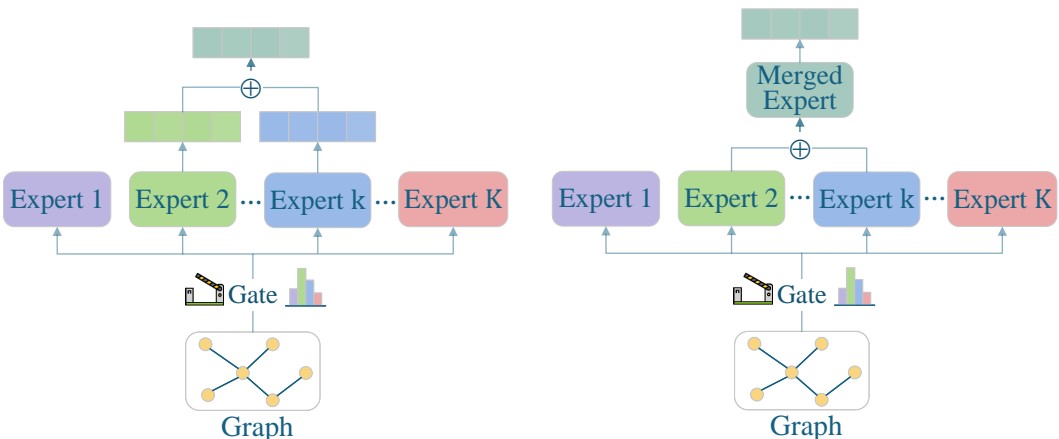

Figure 8: Top-$k$ expert ensembling (expert outputs are averaged).

Figure 9: Standard expert merging (expert parameters are averaged without enhancement).

**Observation 2:** As depicted in Figure 8, we modify the MoE layer of AnyGraph by selecting the top-$k$ experts and aggregating their outputs using a weighted average. Specifically, we replace the top-1 gating in Eq.(5) with a top-$k$ gating mechanism:

$$\mathcal{I} = \text{Top}k\left(\{\psi_i\}_{i=1}^K\right). \tag{25}$$

Here, $\mathcal{I}$ denotes the index set of the top-$k$ experts with the highest competence scores. And the final node embedding matrix in Eq.(6) becomes:

$$\hat{\mathbf{E}} = \sum_{i \in \mathcal{I}} \alpha_i \mathcal{M}_i(\mathbf{E}_1; \theta_i), \quad \alpha_i = \frac{e^{\psi_i}}{\sum_{j \in \mathcal{I}} e^{\psi_j}}, \tag{26}$$

---

[2]Official repository: `https://github.com/HKUDS/AnyGraph`

where $\mathcal{M}_i(\cdot; \theta_i)$ denotes the $i$-th expert, and $\alpha_i$ is its weight, obtained by applying the softmax function to the competence scores of the selected $k$ experts.

**Observation 3:** As demonstrated in Figure 9, to reduce the training and inference costs, we further modify the MoE layer to adopt an expert fusion approach. Concretely, we fuse the parameters of the top-$k$ selected experts into a single expert by replacing Eq.(26) with:

$$\hat{\mathbf{E}} = \mathcal{M}(\mathbf{E}_1; \sum_{i \in \mathcal{I}} \alpha_i \theta_i), \quad \alpha_i = \frac{e^{\psi_i}}{\sum_{j \in \mathcal{I}} e^{\psi_j}}, \tag{27}$$

where $\mathcal{M}$ is the parameter-merged expert.

## C   Training Pipeline and Complexity Analysis

### C.1   Training Pipeline

The training procedures of KDEM and PPEM are presented in Algorithm 1 and Algorithm 2, respectively. In KDEM, the merged expert is used to compute a self-supervised loss, to which a knowledge-distillation term is periodically added, and the parameters are updated based on the combined objective. In PPEM, expert parameters are first updated using the self-supervised loss, and then the top-$k$ experts are periodically updated via an exponential moving average.

---

**Algorithm 1** Training Pipeline of KDEM

---

**Input:** Training graphs $\mathcal{D}$, preprocessed embeddings $\mathbf{E}_1$, expert models $\{\mathcal{M}_k(\cdot; \theta_k)\}_{k=1}^K$, period $T_1$
    of applying knowledge distillation.
1: **for** every epoch **do**
2:     **for** $i$-th batch of triplets $\mathcal{B} = \{(v_{a_b}, v_{p_b}, v_{n_b})\}$ **do**
3:         Get the indices $\mathcal{I}$ and weights $\alpha$ of the top-$k$ experts via Eq.(7)
4:         Merge the parameters of the top-$k$ experts
5:         Get $\mathbf{E}_1$ and compute $\hat{\mathbf{E}}$ using the merged expert (Eq.(8))
6:         Compute self-supervised loss $\mathcal{L}_{ce}$ using $\hat{\mathbf{E}}$ (Eq.(9))
7:         **if** $(i + 1)$ is divisible by $T_1$ **then**
8:             Compute $\mathcal{L}_{kd}$ (Eq.(10))
9:         **end if**
10:        Compute the final loss (Eq.(11))
11:        Update parameters $\{\theta_i\}_{i \in \mathcal{I}}$ via backpropagation
12:     **end for**
13: **end for**
14: **return** Expert parameters $\{\theta_k\}_{k=1}^K$

---

### C.2   Complexity Analysis

The initial embedding computation and expert routing can be preprocessed, with per-graph time complexities of $\mathcal{O}(|\mathcal{E}| \cdot d \cdot L)$ and $\mathcal{O}(|\mathcal{V}| \cdot d^2 \cdot L' \cdot K + S \cdot d \cdot K)$, respectively, where $L$ and $L'$ denote the number of message passing and MLP layers. Next, we analyze the per-batch time complexity of the MoE module. Both our method and AnyGraph process data using one expert (merged expert vs. top-1), with a time complexity of $\mathcal{O}(|\mathcal{B}| \cdot d^2 \cdot L')$. In addition, our methods incorporate expert merging, which involves averaging of the expert parameters with a cost of $\mathcal{O}(k \cdot d^2 \cdot L')$, where $k \ll |\mathcal{B}|$. KDEM periodically applies expert ensembling, and when averaged over each batch, the time complexity is $\mathcal{O}(\frac{k}{T_1}|\mathcal{B}| \cdot d^2 \cdot L')$, where $T_1$ is the period. In our experiments, $T_1$ is set to 100. PPEM periodically applies an exponential moving average, introducing a time complexity of $\mathcal{O}(\frac{k}{T_2} \cdot d^2 \cdot L')$. Note that during inference, our methods require expert merging only once per dataset, making the associated computational cost practically negligible.

---
**Algorithm 2** Training Pipeline of PPEM
---
**Input:** Training graphs $\mathcal{D}$, preprocessed embeddings $\mathbf{E_1}$, expert models $\{\mathcal{M}_k(\cdot; \theta_k)\}_{k=1}^{K}$, period $T_2$
   of exponential moving average (EMA).
1: **for** every epoch **do**
2:     **for** $i$-th batch of triplets $\mathcal{B} = \{(v_{a_b}, v_{p_b}, v_{n_b})\}$ **do**
3:         Get the indices $\mathcal{I}$ and weights $\alpha$ of the top-$k$ experts via Eq.(7)
4:         Merge the parameters of the top-$k$ experts
5:         Get $\mathbf{E}_1$ and compute $\hat{\mathbf{E}}$ using the merged expert (Eq.(8))
6:         Compute self-supervised loss $\mathcal{L}_{ce}$ using $\hat{\mathbf{E}}$ (Eq.(9))
7:         Update expert parameters $\{\theta_i\}_{i \in \mathcal{I}}$ via backpropagation
8:         **if** $(i+1)$ is divisible by $T_2$ **then**
9:             Update expert parameters $\{\theta_i\}_{i \in \mathcal{I}}$ via EMA (Eq.(13))
10:         **end if**
11:     **end for**
12: **end for**
13: **return** Expert parameters $\{\theta_k\}_{k=1}^{K}$
---

# D  Datasets

Our study leverages an extensive collection of graph datasets drawn from diverse domains, comprising a total of 14,437,372 nodes and 199,265,688 edges. All datasets in this study are obtained from prior research [33, 21, 40]. The dataset statistics are presented in Table 5. These datasets are organized into multiple categories and groups to address specific research questions and evaluation needs. All datasets are used for link prediction, five additional datasets are also used for node classification.

## D.1  Dataset Categories

**1. E-commerce Datasets**  This category includes 13 datasets extracted from various e-commerce contexts such as online retail services and user-rating platforms. While these datasets display variability in node feature availability and generation, some share common methods. For example, Amazon-text, Steam-text, and Yelp-text employ one feature generation strategy, whereas Fitness, Photo, and Goodreads consistently apply an alternative method.

**2. Academic Network Datasets**  This category encompasses 9 datasets that model scholarly interactions such as paper citations and author collaborations across diverse research disciplines. The datasets vary in feature construction methods, employing techniques like bag-of-words models, NLP-based embeddings, and features derived from large language models. Datasets in this category include Cora, PubMed, Citeseer, Arxiv, Arxiv-t (which uses a unique feature derivation method), CS, Citation-2019, Citation-20Century, and OGB-Collab.

**3. Biological Information Networks**  This set comprises six datasets centered on biological entities such as proteins, drugs, and diseases. Specifically, this group includes the OGB-DDI and OGB-PPA networks, along with four species-specific protein relation datasets designated Proteins-0, Proteins-1, Proteins-2, and Proteins-3.

**4. Miscellaneous Datasets**  In addition to the primary domains, we incorporate 5 datasets originating from distinct and less thematically consistent domains. These datasets provide additional diversity and include: the Email-Enron network (emails), Web-Stanford (website connectivity), RoadNet-PA (road networks), P2P-Gnutella06 (peer-to-peer web), and Soc-Epinions1 (trust networks).

## D.2  Dataset Grouping for Performance Evaluation

To streamline performance comparisons and effectively prevent information leakage, particularly in zero-shot settings, the datasets are further organized into groups. This grouping is based on their respective sources, feature generation methods, and specific domains, as detailed below:

Table 5: Statistics of the datasets. All datasets are used for link prediction, and five additional datasets are also used for node classification (those marked with a check in the last column).

| Datasets | # Nodes | # Edges | # Feats | Groups | Node Cls. |
|---|---|---|---|---|---|
| DDI | 4,267 | 1,334,889 | 0 | Link2 / Others | |
| Collab | 235,868 | 1,285,465 | 128 | Link2 / Academic | |
| ML1m | 9,746 | 920,193 | 0 | Link2 / E-commerce | |
| ML10m | 80,555 | 9,200,050 | 0 | Link2 / E-commerce | |
| Amazon-book | 144,242 | 2,984,108 | 0 | Link1 / E-commerce | |
| PPA | 576,289 | 45,495,642 | 58 | Link1 / Others | |
| Yelp2018 | 69,716 | 1,561,406 | 0 | Link1 / E-commerce | |
| Gowalla | 70,839 | 1,027,370 | 0 | Link2 / E-commerce | |
| Cora | 2,708 | 10,556 | 1433 | Link2 / Academic | ✓ |
| PubMed | 19,717 | 88,648 | 500 | Link1 / Academic | ✓ |
| Citeseer | 3,327 | 9,104 | 3703 | Link1 / Academic | |
| Proteins-0 | 25,449 | 11,660,646 | 0 | Link2 / Others | |
| Proteins-1 | 6,568 | 1,845,960 | 0 | Link2 / Others | |
| Proteins-2 | 18,108 | 7,418,688 | 0 | Link2 / Others | |
| Proteins-3 | 13,015 | 3,962,930 | 0 | Link2 / Others | |
| Products-home | 9,790 | 131,843 | 100 | Link1 / E-commerce | ✓ |
| Products-tech | 47,428 | 2,077,241 | 100 | Link1 / E-commerce | ✓ |
| Yelp-t | 22,101 | 277,535 | 1536 | Link1 / E-commerce | |
| Amazon-t | 20,332 | 200,860 | 1536 | Link1 / E-commerce | |
| Steam-t | 28,547 | 525,922 | 1536 | Link1 / E-commerce | |
| Goodreads | 676,084 | 8,582,306 | 768 | Link2 / E-commerce | |
| Fitness | 173,055 | 1,773,500 | 768 | Link2 / E-commerce | |
| Soc-Epinions1 | 75,879 | 508,837 | 0 | Link1 / Others | |
| Email-Enron | 36,692 | 183,831 | 0 | Link1 / Others | |
| Web-Stanford | 281,903 | 2,312,497 | 0 | Link2 / Others | |
| RoadNet-PA | 1,088,092 | 1,541,898 | 0 | Link2 / Others | |
| P2P-Gnutella | 8,717 | 31,525 | 128 | Link1 / Others | |
| Citation-2019 | 765,658 | 1,917,381 | 128 | Link1 / Academic | |
| Citation-20Century | 1,016,241 | 5,565,798 | 128 | Link1 / Academic | |
| Arxiv | 169,343 | 1,166,243 | 128 | Link2 / Academic | ✓ |
| Arxiv-t | 169,343 | 1,166,243 | 768 | Link2 / Academic | |
| Photo | 48,362 | 500,939 | 768 | Link2 / E-commerce | |
| CS | 18,333 | 163,788 | 6805 | Link2 / Academic | |

**1. Balanced Groups:** Two primary groups, *Link1* and *Link2*, are formed exclusively from link prediction datasets. The total number of edges in each group is approximately equal, and within each domain, the edge counts of the individual datasets are likewise comparable. Datasets collected from the same source (e.g., Movielens-1M and Movielens-10M) or those sharing identical feature generation methods (e.g., Fitness and Photo) are kept together to ensure evaluation integrity.

- *Link1* **Group:** Contains 15 datasets such as Products-tech, Yelp2018, Yelp-textfeat, Products-home, Steam-text, Amazon-text, Amazon-book, Citation-2019, Citation-20Century, PubMed, Citeseer, OGB-PPA, P2P-Gnutella06, Soc-Epinions1, and Email-Enron.

- *Link2* **Group:** Comprises 18 datasets, i.e., Photo, Goodreads, Fitness, Movielens-1M, Movielens-10M, Gowalla, Arxiv, Arxiv-t, Cora, CS, OGB-Collab, Proteins-0, Proteins-1, Proteins-2, Proteins-3, OGB-DDI, Web-Stanford, and RoadNet-PA.

**2. Domain-Specific Groups:** To enable a more focused analysis of model performance and generalization capabilities across distinct thematic areas, we define domain-specific groups by consolidating datasets based on their primary domain of origin or application. This categorization allows for a targeted assessment of how models perform, adapt, or exhibit specific behaviors within these coherent operational contexts, revealing domain-specific strengths or limitations:

- *E-commerce* **Group:** Containing all datasets from the e-commerce categories.

- *Academic* **Group:** Containing all datasets from the academic categories.
- *Others* **Group:** This comprises all biological datasets along with the other miscellaneous datasets such as Email-Enron, Web-Stanford, RoadNet-PA, P2P-Gnutella06, and Soc-Epinions1.

# E   Experimental details

## E.1   Baselines

We benchmark our approaches against eleven representative baselines spanning four distinct categories: recent graph foundation models, graph prompt learning methods, self-supervised pre-training methods, and classic graph neural networks.

**Graph Foundation Models**   We adopt six recently proposed GFMs, which are described as follows:

- OpenGraph [41] is designed to generalize across diverse and unseen graph data by integrating a unified graph tokenizer, a scalable graph transformer, and LLM-enhanced data augmentation. This architecture enables strong zero-shot learning performance by capturing global topological patterns and adapting to varying graph properties without requiring retraining.
- GraphGPT [33] integrates LLMs with graph structural knowledge through a graph instruction tuning paradigm. It employs a dual-stage instruction tuning process—starting with self-supervised graph matching to align graph structures with natural language, followed by task-specific fine-tuning—to enhance the LLM's ability to understand and reason over graph data, achieving strong performance in both supervised and zero-shot graph learning tasks.
- AnyGraph [40] is a versatile GFM designed to address the challenges of structure and feature heterogeneity across diverse graph datasets. Built upon an MoE architecture, it incorporates a dynamic expert routing mechanism that enables efficient adaptation to new graph domains and exhibits scaling law behavior, where performance improves with increased data and model size.
- UniGraph [13] is a unified foundation model for text-attributed graphs, which employs a novel cascaded architecture of language models and GNNs alongside a self-supervised pre-training algorithm based on masked graph modeling and instruction tuning.
- GFT [37] treats computation trees as transferable patterns and learns a graph vocabulary by performing several computation tree reconstruction tasks, encoding general graph knowledge into the vocabulary, which can then be adapted to downstream tasks through fine-tuning.
- GOFA [19] presents a novel generative foundation model for graph-language tasks by interleaving trainable GNN layers into a frozen pre-trained LLM compressor, thereby marrying structural graph understanding with free-form text generation.

**Graph Prompt Learning Methods**   This category includes two prompt-based tuning approaches:

- GraphPrompt [25] unifies pre-training and downstream tasks into a common task template by employing a learnable prompt. This assists downstream tasks in locating the most relevant knowledge from the pre-trained model in a task-specific manner.
- GPF [6] is a universal prompt-based tuning method for pre-trained GNN models under any pre-training strategy. It operates on the input graph's feature space and can theoretically achieve an equivalent effect to any form of prompting function, eliminating the need to design specific prompting functions for each pre-training strategy.

**Graph Self-supervised Pre-training Methods**   We employ GraphCL [46] as a representative method. GraphCL maximizes agreement between differently augmented views of the same graph, enabling effective pretraining without labels. It introduces a set of graph-specific augmentations, such as node dropping and edge perturbation, to generate diverse yet semantically consistent views for contrastive learning.

**Graph Neural Networks**   This category includes two classic GNNs: Graph Attention Network (GAT [34]) and Graph Isomorphism Network (GIN [42]). GAT leverages attention mechanisms to assign different importances to neighboring nodes, while GIN is designed to maximally preserve graph structure by mimicking the power of the Weisfeiler-Lehman test for graph isomorphism.

### E.2 Evaluation Protocols

We follow the same dataset splits as prior works [33, 21, 40]. Our method adopts the same zero-shot setting as AnyGraph. We train two separate models on *Link1* and *Link2* respectively, then evaluate the zero-shot performance of the *Link1*-trained model on *Link2* and vice versa. Models are cross-applied between *Link1* and *Link2*'s domain-specific data. For example, we apply the model trained on *Link1* to the Academic graphs from *Link2*, and conversely apply the *Link2*-trained model to *Link1*'s Academic data. The results are then aggregated through a weighted average based on the number of edges in each dataset. For the six graph foundation models, we either report the results from their original papers or evaluate them using their officially released code. GAT and GIN are trained from scratch under a few-shot setting (with 10% training samples). GraphCL, GraphPrompt, and GPF undergo pretraining followed by fine-tuning on the evaluation datasets.

**Zero-shot Setting for Node Classification** In our zero-shot node classification framework, we represent label classes as distinct nodes and connect nodes with training labels to their corresponding class nodes. This method eliminates the need for learning separate parameters for each class, simplifying the zero-shot learning process. By integrating this method into baseline models, we enhance their ability to efficiently handle unseen node labels.

**Evaluation Metrics** For link prediction, we follow previous work [40] and use Recall@20 and NDCG@20 as evaluation metrics. For node classification, we use accuracy and macro F1 as metrics. The results for each group of datasets are averaged based on the number of test samples.

### E.3 Implementation Details and Hyperparameters

All experiments are conducted on a single NVIDIA GeForce RTX 3090 GPU (24GB VRAM). Our methods are implemented using PyTorch, where SVD is computed via the built-in `svd_lowrank` function for enhanced computational efficiency. When the number of training edges in a large dataset exceeds $500\times$ the batch size, we randomly sample 500 batches from it in each epoch. The models are trained for 100 epochs using the Adam optimizer with a batch size of 4096 and a learning rate of either $1 \times 10^{-4}$ (*Link1*) or $2 \times 10^{-4}$ (*Link2*). The number of message passing layers $L$ is set to 3. We use 8 expert models, each consisting of an 8-layer MLP with 512 units per layer. The training step interval $T_1$ for knowledge distillation is set to 100, with a loss weight $\gamma$=0.01. The training step interval $T_2$ for the exponential moving average (Eq.(13)) is selected from {10,20,50,100,200}, with the decay factor $\beta$ chosen from {0.99,0.999,0.9995,0.9999}. The recalibration strength parameter $\rho$ (Eq.(23)) is set to 0.2.

## F Fine-grained Results

We present the fine-grained link prediction results for each dataset in Table 6. We evaluate AnyGraph, the standard expert fusion method (Figure 9), and our enhanced expert fusion method under both full-shot (train on *Link1*, test on *Link1*) and zero-shot (train on *Link1*, test on *Link2*) settings. As shown, AnyGraph performs well under the full-shot setting but struggles in the more common real-world zero-shot scenario. This may be due to its reliance on a single expert, which makes it susceptible to overfitting on the training set. In contrast, all expert merging methods leverage the knowledge of multiple experts, outperforming AnyGraph in the zero-shot setting. Moreover, our methods achieve the best overall performance, surpassing both AnyGraph and vanilla expert merging strategies, thereby demonstrating the superior generalization ability of our approaches.

The fine-grained node classification results compared to AnyGraph are shown in Table 7. Overall, KDEM and PPEM achieve higher macro-F1 scores on 4 out of 5 datasets. Specifically, KDEM improves the average macro-F1 by 1.79% (from 43.24 to 45.03), while PPEM increases it by 0.92% (to 44.16), although both methods show slight decreases in overall accuracy relative to the baseline. Notably, on the Arxiv dataset, KDEM's macro-F1 rises substantially from 36.50 to 41.76, underscoring the significant advantage of multi-expert fusion strategies in addressing the class imbalance issue. Given the large variance in class sizes in the Arxiv test set (ranging from dozens to over ten thousand), the expert fusion strategy effectively mitigates the challenges posed by skewed class distributions by enhancing discrimination of minority classes. This is because the multi-expert

Table 6: Fine-grained link prediction results for each dataset, compared with AnyGraph and standard/vanilla expert merging (Figure 9). Vanilla EM refers to the vanilla expert merging method, which directly computes a weighted average of the parameters from the top-$k$ experts without any additional enhancements. All models are trained on *Link1* and then tested on *Link1* (full-shot) and *Link2* (zero-shot), respectively. The results for each dataset are averaged over 5 runs. The averages over dataset groups are computed by weighting each dataset according to the number of test edges.

| Groups | Datasets | AnyGraph | | Vanilla EM | | KDEM | | PPEM | |
|---|---|---|---|---|---|---|---|---|---|
| | | Recall | NDCG | Recall | NDCG | Recall | NDCG | Recall | NDCG |
| Link1 full-shot | Amazon-book | 4.39 | 3.53 | 4.41 | 3.51 | 4.38 | 3.50 | 4.47 | 3.56 |
| | Amazon-t | 13.74 | 9.12 | 12.25 | 8.18 | 12.58 | 8.32 | 13.04 | 8.55 |
| | Citation-2019 | 13.76 | 6.05 | 14.49 | 6.65 | 15.40 | 7.10 | 14.04 | 6.29 |
| | Citation-20C | 35.01 | 14.74 | 30.09 | 12.28 | 31.66 | 12.76 | 34.59 | 14.71 |
| | Citeseer | 79.34 | 64.89 | 79.24 | 62.06 | 79.31 | 65.85 | 79.47 | 65.94 |
| | Email-Enron | 64.43 | 44.50 | 64.93 | 44.83 | 64.15 | 44.70 | 64.52 | 44.86 |
| | P2P-Gnutella | 4.72 | 1.98 | 3.50 | 1.22 | 3.41 | 1.21 | 3.61 | 1.33 |
| | PPA | 25.64 | 14.49 | 24.22 | 13.56 | 25.65 | 14.43 | 23.96 | 13.46 |
| | Products-home | 70.63 | 42.09 | 69.87 | 40.82 | 69.39 | 41.31 | 69.29 | 41.84 |
| | Products-tech | 52.42 | 34.09 | 51.50 | 34.14 | 50.49 | 33.33 | 52.32 | 34.52 |
| | PubMed | 76.15 | 68.88 | 74.99 | 62.92 | 74.86 | 64.11 | 76.27 | 68.70 |
| | Soc-Epinions1 | 24.86 | 15.19 | 24.57 | 14.96 | 24.71 | 15.26 | 24.68 | 15.26 |
| | Steam-t | 10.63 | 6.63 | 8.79 | 5.44 | 9.79 | 6.13 | 9.21 | 5.70 |
| | Yelp2018 | 5.08 | 4.17 | 4.94 | 4.05 | 5.03 | 4.13 | 5.00 | 4.12 |
| | Yelp-t | 11.01 | 6.92 | 7.17 | 4.46 | 8.06 | 5.09 | 8.13 | 5.07 |
| | **Average** | **27.34** | **14.09** | **25.03** | **12.85** | **26.20** | **13.40** | **26.45** | **13.65** |
| Link2 zero-shot | CS | 79.85 | 63.54 | 79.14 | 62.06 | 79.59 | 63.78 | 79.18 | 63.88 |
| | Fitness | 45.59 | 24.95 | 52.73 | 29.96 | 55.74 | 33.47 | 55.50 | 33.14 |
| | Goodreads | 27.61 | 14.88 | 32.38 | 17.34 | 40.23 | 23.51 | 38.72 | 22.43 |
| | Photo | 47.14 | 25.36 | 54.93 | 30.33 | 57.79 | 34.07 | 56.53 | 33.45 |
| | Arxiv | 36.91 | 16.96 | 34.75 | 15.77 | 35.63 | 16.59 | 34.88 | 15.85 |
| | Arxiv-t | 38.67 | 18.58 | 35.19 | 16.47 | 39.71 | 20.37 | 39.21 | 20.32 |
| | Collab | 3.66 | 1.76 | 3.80 | 1.81 | 3.46 | 1.68 | 3.60 | 1.74 |
| | Cora | 82.19 | 65.87 | 81.83 | 67.74 | 81.73 | 66.16 | 81.13 | 65.27 |
| | DDI | 7.66 | 21.01 | 9.38 | 19.92 | 8.45 | 16.32 | 9.39 | 19.67 |
| | Gowalla | 11.18 | 8.06 | 9.66 | 6.92 | 12.57 | 9.26 | 10.48 | 7.56 |
| | ML10m | 6.51 | 8.56 | 24.11 | 28.00 | 23.40 | 27.18 | 19.52 | 23.66 |
| | ML1m | 10.65 | 16.43 | 13.35 | 20.15 | 11.41 | 17.78 | 10.50 | 16.32 |
| | Proteins-0 | 16.93 | 23.23 | 17.30 | 23.73 | 17.57 | 23.99 | 18.81 | 25.82 |
| | Proteins-1 | 19.68 | 24.05 | 23.64 | 27.78 | 22.04 | 25.93 | 25.77 | 29.46 |
| | Proteins-2 | 14.02 | 20.07 | 17.80 | 24.95 | 17.84 | 25.27 | 13.66 | 20.22 |
| | Proteins-3 | 23.94 | 26.37 | 23.24 | 26.02 | 21.92 | 25.26 | 23.05 | 26.18 |
| | RoadNet-PA | 89.47 | 50.42 | 89.15 | 52.62 | 88.83 | 52.93 | 88.91 | 52.83 |
| | Web-Stanford | 72.39 | 52.02 | 71.67 | 51.79 | 72.06 | 52.28 | 71.83 | 52.78 |
| | **Average** | **45.43**[†] | **27.01**[†] | **48.34** | **29.72** | **51.69** | **32.60** | **50.77** | **31.93** |

[†] Since AnyGraph does not report results for each individual dataset, we directly evaluate the released model and present the results. Therefore, the overall zero-shot average we report may differ slightly from that in the original paper (45.43 vs. 46.42 for recall, and 27.01 vs. 27.21 for NDCG).

mechanism integrates diverse perspectives from different experts, leveraging their diversity to reduce the risk of errors arising from a single expert's bias.

Additional node classification results compared with other GFMs are displayed in Table 8. These models tap into the robust generalization abilities of large language models (LLMs), pretrained on massive text corpora, to elevate their performance in graph learning. For example, OpenGraph employs LLMs for data augmentation, while other methods leverage them to encode textual node attributes or predict node classes directly. The results reveal a striking trend: LLM-based GFMs deliver superior accuracy on small-scale datasets with fewer classes, such as Cora (7 classes), yet their performance in zero-shot and few-shot scenarios falters on larger, class-rich datasets like Arxiv (40 classes). This drop-off likely stems from the heightened complexity of processing lengthy contexts and intricate graph structures, which taxes the models' reasoning and generalization strengths.

Table 7: Fine-grained node classification results, compared with AnyGraph. The results for each dataset are averaged over 10 runs. The overall average (last row) is computed by weighting each dataset according to the number of test nodes.

| Datasets | AnyGraph | | KDEM | | PPEM | |
|---|---|---|---|---|---|---|
| | Acc | Macro-F1 | Acc | Macro-F1 | Acc | Macro-F1 |
| Arxiv | 62.05 | 36.50 | 61.86 | 41.76 | 61.20 | 39.09 |
| Cora | 62.27 | 56.00 | 62.45 | 56.32 | 62.10 | 55.93 |
| Products-home | 66.36 | 40.31 | 65.44 | 41.37 | 67.30 | 48.03 |
| PubMed | 69.80 | 67.49 | 70.36 | 68.83 | 69.94 | 67.73 |
| Products-tech | 74.45 | 64.89 | 69.42 | 48.56 | 66.31 | 55.30 |
| **Average** | 64.38 | 43.24 | 63.56 | 45.03 | 62.65 | 44.16 |

Table 8: Fine-grained node classification results, compared with other existing GFMs (accuracy). The results of GFT are under a few-shot setting. The Arxiv results of GraphGPT and UniGraph are under supervised learning and few-shot settings, respectively.

| | GraphGPT | OpenGraph | UniGraph | GFT | GOFA | KDEM | PPEM |
|---|---|---|---|---|---|---|---|
| Cora | 18.13 | 75.04 | 69.53 | 67.36 | 70.81 | 62.45 | 62.10 |
| PubMed | 70.11 | 68.69 | 72.48 | - | - | 70.36 | 69.94 |
| Arxiv | 62.58 | - | 31.35 | 36.29 | - | 61.86 | 61.20 |

## G  Additional Experimental Results

### G.1  Comparison with the Best Single Expert

We demonstrate in Table 9 that both of our enhanced expert merging strategies, KDEM and PPEM, outperform the best individual expert. This improvement arises because our methods effectively integrate knowledge from multiple experts, thereby reducing the potential bias or overfitting of any single expert. Notably, although Figure 7 shows that the second-ranked expert (as assigned by the router) performs worse than the top-ranked expert in isolation, merging their parameters still leads to superior overall performance. This is because even lower-ranked experts can capture complementary patterns or task-relevant nuances that the top expert might miss. By combining their knowledge, the model can cover a broader set of features or relationships, improving robustness and generalization. Furthermore, the ensemble effect helps smooth out individual expert errors, allowing the fused model to achieve better aggregate predictions than any one expert alone.

Table 9: Performance comparison with the single top-performed expert. On the *Link1* group, the top-ranked expert achieves the best performance, while on the *Link2* group, the third-ranked expert performs best.

| Method | Link1 | | Link2 | |
|---|---|---|---|---|
| | Recall | NDCG | Recall | NDCG |
| Top-performed expert | 23.94 | 12.68 | 48.22 | 30.56 |
| KDEM | 24.11 | 12.80 | 51.69 | 32.60 |
| PPEM | 24.33 | 12.93 | 50.77 | 31.93 |

### G.2  Parameter Sensitivity Analysis

This section studies the impact of key hyperparameters on our models. Based on the results presented in Figure 10, our observations are as follows:

(1) Selecting multiple experts is essential. As shown in the first subfigure, integrating multiple experts significantly enhances model performance. This suggests that the fused experts can leverage complementary expertise, thereby improving generalization ability. However, using too many experts

can lead to suboptimal performance, as lower-ranked experts may not be well-suited for the given graph data, and merging their parameters can cause knowledge conflicts.

(2) While knowledge distillation applied at each training step ($T_1 = 1$) leads to satisfactory performance, it will incur significant computational cost. Periodic application of knowledge distillation can achieve comparable even better performance, similar to real-world scenarios where teachers do not need to teach students every day.

(3) The experimental results for KDEM are relatively insensitive to the weight of the knowledge distillation loss, $\gamma$, as long as it is not too small. Empirically, a value of 0.01 works well.

(4) The period $T_2$ and decay factor $\beta$ of the EMA need to be appropriately chosen. Smaller values for $T_2$ and $\beta$ lead to faster changes in expert parameters, which may cause the experts to become too similar, preventing the merged expert from effectively utilizing the diversity of knowledge. This confirms that experts should retain a degree of diversity while maintaining some level of similarity, so the fused expert can benefit from the expertise of all contributors.

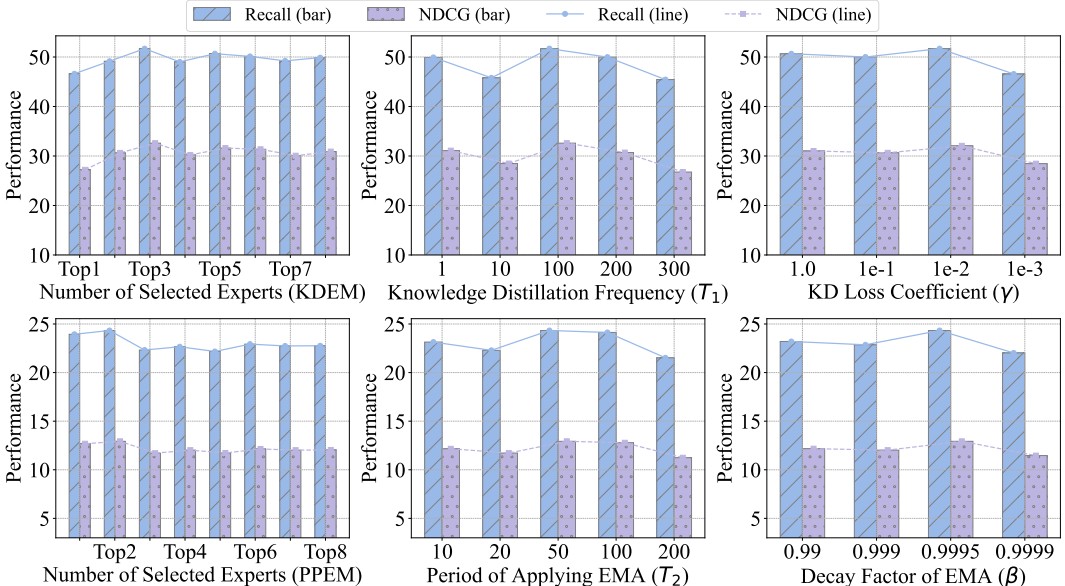

Figure 10: Parameter Sensitivity Analysis. The first row shows the test results of the KDEM model trained on the *Link1* group and evaluated on the *Link2* group, while the second row shows the test results of the PPEM model trained on the *Link2* group and evaluated on the *Link1* group.

### G.3 Effect of the Number of Experts

This section investigates the impact of the number of experts on model performance. As presented in Table 10, the results show that increasing the total number of experts from 2 to 8 yields consistent gains in both recall and NDCG, with the most pronounced improvements on *Link2*.

Table 10: Performance of KDEM with varying numbers of experts and activated experts.

| # Experts | # Activated Experts | Link1 | | Link2 | |
|:---:|:---:|:---:|:---:|:---:|:---:|
| | | Recall | NDCG | Recall | NDCG |
| 2 | 1 | 22.85 | 12.02 | 45.68 | 27.64 |
| 4 | 1 | 22.23 | 11.72 | 48.02 | 29.31 |
| 4 | 2 | 21.53 | 11.13 | 48.64 | 29.43 |
| 4 | 3 | 22.66 | 11.97 | 48.02 | 29.31 |
| 8 | 2 | 24.11 | 12.80 | 49.15 | 30.63 |
| 8 | 3 | 23.35 | 12.33 | 51.69 | 32.60 |

# H  Further Discussion of the Expert Assignment Mechanism

To gain a comprehensive understanding of the routing mechanism, we record the expert assignment behavior of a trained PPEM model (trained on the *Link2* group and evaluated on the *Link1* group), and plot the competence scores of all 8 experts across all datasets, as shown in Figure 11. From this, we derive the following observations and insights.

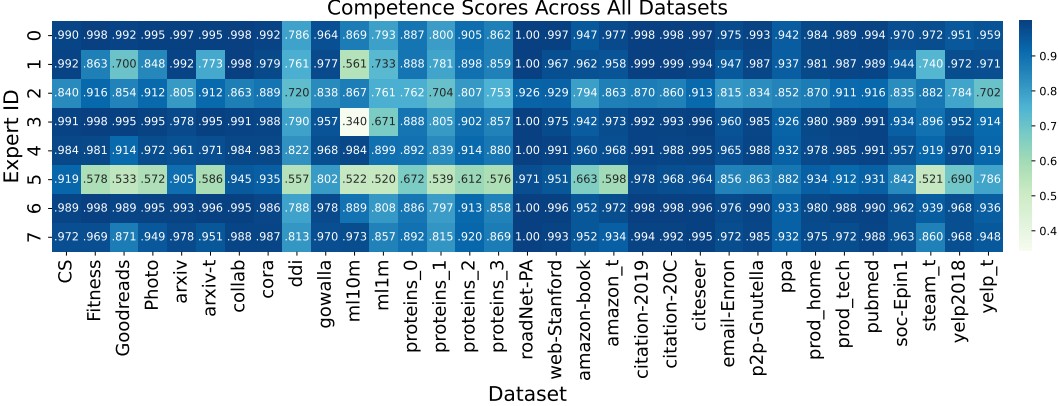

Figure 11: Competence scores of 8 experts across all datasets. These expert scores are obtained from a PPEM model trained on the *Link2* set and evaluated on the *Link1* group. The left half of the figure corresponds to the training datasets (from CS to Web-Stanford), while the right half represents the test datasets (from Amazon-book to Yelp-t).

## H.1  Transferability of the Routing Mechanism

We observe that the routing mechanism exhibits strong transferability from the pretraining data to previously unseen test data. For example, in the training set (*Link2*), Experts 0, 3, and 6 consistently specialize in e-commerce datasets such as Photo, Goodreads, and Fitness. Similarly, in the test set (*Link1*), the e-commerce dataset Products-Tech is assigned the highest routing scores to these same experts, indicating that the router effectively generalizes expert assignment patterns across domains.

## H.2  Additional Insights

We also observe that Expert 2 and Expert 5 are rarely selected. To understand the root cause, we analyze the model's training dynamics and find that these two experts receive minimal updates, leading to a training imbalance. A similar phenomenon is observed in AnyGraph, which employs the same routing mechanism, suggesting that this issue is not unique to our model. We hypothesize that the limited diversity in the training set–comprising only 18 graphs–results in a small number of structural clusters, reducing the opportunity for all experts to be engaged.

Table 11: Results of changing the calibration parameter $\rho$. Increasing $\rho$ enforces stricter expert balancing but leads to a drop in performance.

| $\rho$ | 0 | 0.1 | 0.2 | 0.3 | 0.4 |
|---|---|---|---|---|---|
| Balanced | ✗ | ✗ | ✗ | ✓ | ✓ |
| Recall | 48.20 | 48.35 | 50.77 | 49.11 | 49.84 |
| NDCG | 30.07 | 29.83 | 31.93 | 30.08 | 30.47 |

To encourage more balanced expert utilization, we adjust the calibration range parameter $\rho$ (Eq.(23)). Although increasing this parameter leads to a more uniform expert assignment, as shown in Table 11, it results in a drop in overall model performance. This trade-off is consistent with findings from prior work [36, 43], which suggest that enforcing balanced routing may result in suboptimal expert selection. Notably, in LLMs based on the MoE paradigm, load balancing is typically adopted not to improve performance, but to accelerate training and inference through parallelization strategies.

The load balancing strategies are not the main focus of this work and fall outside the scope of our study. Potential directions for future work include exploring auxiliary-loss-free load balancing methods [36] or incorporating expert pruning strategies [26].

### H.3   Change the Routing Mechanism

The routing mechanism we adopt is non-parametric; to more comprehensively investigate the impact of routing strategies, we replace it with a trainable, parameterized router commonly used in LLMs [7]. Specifically, we average-pool the representations $\mathbf{E_1}$ (Eq.3) and feed them into a linear router to compute an affinity score for each expert, then select the top-$k$ experts with the highest scores for fusion. In addition, we also evaluate the expert choice router [52] as an alternative routing strategy. As reported in Table 12, using these trainable, parameterized routers yields inferior results—likely because the heuristic non-parametric router more accurately assigns structurally similar graphs to the same experts, whereas the learnable router can introduce instability during training [52, 29].

Table 12: Effect of changing the router to commonly used trainable ones.

| Method | Link1 | | Link2 | |
|---|---|---|---|---|
| | Recall | NDCG | Recall | NDCG |
| PPEM w/ standard trainable router | 22.19 | 11.82 | 50.25 | 31.38 |
| PPEM w/ expert choice routing | - | - | 50.06 | 30.91 |
| PPEM | **24.33** | **12.93** | **50.77** | **31.93** |

## I   Deeper Discussions of the Two Approaches

The two enhanced expert merging strategies, KDEM (Section 4.1) and PPEM (Section 4.2), though distinct in their mechanics—KDEM operating in the output space and PPEM in the parameter space—share common characteristics and offer insightful perspectives on optimizing the MoE layers.

**Commonalities of KDEM and PPEM**   **(1) Alignment with Expert Ensemble Performance.** Both methodologies address the observed performance degradation that can occur with standard expert merging techniques and fundamentally strive to elevate the performance of expert merging to levels comparable with more computationally intensive yet effective expert ensembling. While KDEM leverages knowledge distillation to directly align the merged expert's outputs with those of the ensemble, PPEM exploits a theoretical insight to approximate ensemble behavior by bringing expert parameters closer via EMA. **(2) Computational Efficiency.** Efficiency remains a core tenet for both, as they aim to sidestep the substantial computational demands of direct expert ensembling. KDEM introduces a modest additional computational step through periodic ensemble computation, whereas PPEM is engineered to deliver comparable outcomes with negligible overhead by operating directly within the parameter space. **(3) Regularization Effect.** KDEM explicitly introduces a regularization effect via its distillation mechanism, guiding the merged expert toward the "soft target" distribution of the ensemble. This encourages the merged expert to adopt a broader, more generalizable representation. PPEM, by fostering parameter similarity among selected experts, may also confer an implicit regularization effect. This can prevent individual experts from diverging too dramatically, potentially improving model robustness and generalization by reducing interference during the merge.

**Insights Gained from the Two Methods**   **(1) Flexibility in Design.** The existence of two viable strategies highlights the flexibility in designing expert merging techniques. KDEM's output-space approach leverages a well-established technique (knowledge distillation), making it intuitive and adaptable to various MoE setups. PPEM's parameter-space method, rooted in theoretical principles, offers a computationally lean alternative that requires minimal overhead. This versatility suggests that future methods could explore hybrid approaches or context-specific adaptations based on dataset characteristics or resource constraints. **(2) Trade-Offs and Practical Considerations.** KDEM offers robust performance improvements but requires periodic ensemble computations, adding moderate training cost. It may be preferable when computational resources allow for this overhead and high

accuracy is paramount. PPEM achieves comparable results with negligible cost, making it ideal for resource-constrained scenarios. The choice between KDEM and PPEM could be tailored to specific applications, such as prioritizing efficiency in large-scale e-commerce systems or accuracy in academic graph analysis. **(3) Potential for Hybridization.** The success of two different approaches indicates that they may possess complementary advantages and potentially offers synergistic benefits if combined. Future work could explore this synergy to fully release potential.

## J  Extended Related Work

**Model Merging and Expert Merging**   Model merging integrates parameters from multiple networks to harness their complementary strengths (see surveys [45, 20]). Expert merging—its counterpart in MoE architectures—combines only specialized experts rather than entire models, yielding two key benefits: reduced compute costs and fully differentiable routing via soft fusion. For example, MEO [11] averages the parameters of the top-$k$ experts to cut overhead, SMEAR [28] fuses all experts to avoid discrete router optimization (and outperforms gradient-estimation baselines), and Lory [51] generalizes SMEAR's principles to large autoregressive LMs. Nonetheless, existing merging strategies remain relatively rudimentary and do not fully realize the potential of combining multiple experts' knowledge.

**Mixture-of-Experts (MoE) on Graphs**   With the widespread application of MoE in the NLP domain [2], they have recently been adopted in graph learning as well. For example, GraphDIVE [16] utilizes MoE to integrate multi-view graph representations, addressing class imbalance issues in graph classification. GMoE [35] treats multiple independent message-passing functions as experts and uses a gating mechanism to assign aggregation experts for each node. MoG [48] dynamically selects sparsification strategies for nodes with tailored sparsifier experts. GraphMETRO [39] employs an MoE architecture to address complex graph distribution shifts, enhancing the generalization ability of graph neural networks. Mowst [47] utilizes a weak MLP expert for features and a strong GNN expert for structure and designs a novel MoE mechanism to integrate their expert knowledge. Node-MoE [10] employs an MoE framework to adaptively select appropriate filter experts for different nodes. GraphAlign assigns different feature transformation experts to each node, aligning the features from different data sources. Despite these advancements, research on MoE in graphs still requires further exploration to fully maximize its potential.

## K  Broader Impact

Our work advances MoE-based GFMs by enabling efficient fusion of multiple experts, leading to improved accuracy and robustness across a variety of graph-related tasks, such as academic network analysis, molecular interaction prediction, and recommendation systems. These improvements have the potential to accelerate discovery in domains like drug development by more effectively modeling molecular interaction graphs and to enhance personalization in e-commerce platforms.

In addition, since MoE layers in large language models are typically composed of MLPs, our proposed expert fusion strategies are readily applicable to this broader class of models. By reducing both training and inference overhead while maintaining performance competitive with top-$k$ expert ensembles, our methods contribute to more sustainable machine learning practices by lowering computational costs and associated carbon emissions. Furthermore, this efficiency paves the way for more practical deployment of LLMs on resource-constrained edge devices.

## L  Limitations

To ensure a fair comparison with AnyGraph and highlight the advantages of our proposed expert merging strategy, we follow its setup by excluding graph-level tasks (e.g., graph classification). We leave a comprehensive investigation of these tasks to future work. Since our approach does not integrate LLMs, it cannot generate free-text responses to broad graph queries as GOFA [19] does; nonetheless, our model demonstrates strong graph understanding and can serve as an effective graph encoder, which, when coupled with an LLM-based decoder and instruction fine-tuning, could achieve comparable capabilities. We leave these extensions to future work.

