# OpenReview forum: "Enhanced Expert Merging for Mixture-of-Experts in Graph Foundation Models"
_NeurIPS.cc/2025/Conference — NeurIPS 2025 poster_

### Official Review · Reviewer_Etth · 2025-06-29

**Clarity:** 4
**Significance:** 3
**Originality:** 3
**Rating:** 5
**Confidence:** 3

**Summary:**

This paper investigates mixture-of-experts (MoE) mechanisms in graph foundation models (GFMs) and addresses the challenge that while GFMs use MoE to handle graph heterogeneity, current approaches only utilize the single top-ranked expert, potentially leaving valuable knowledge from other high-performing experts untapped. Through experimental analysis of AnyGraph, the authors discover that experts ranked second and third by the router often outperform the top-ranked expert, motivating them to explore multi-expert knowledge integration. They propose two enhanced expert merging strategies: (1) Knowledge Distillation Enhanced Expert Merging (KDEM), which uses knowledge distillation to align the behavior of parameter-fused experts with expert ensembles, and (2) Parameter Proximity Enhanced Expert Merging (PPEM), which leverages theoretical analysis showing that when MLP experts have similar parameters, their merged output approximates the ensemble output. The proposed approach bridges the efficiency of expert merging with the performance benefits of expert ensembling through these theoretically-grounded and practically-efficient approaches. Extensive experiments demonstrate that KDEM and PPEM achieve significant performance improvements over the AnyGraph baseline while maintaining computational efficiency close to single-expert inference, which effectively solve the trade-off between performance and efficiency in MoE-based GFMs.

**Questions:**

•	Have the authors investigated whether the non-parametric routing mechanism is actually optimal for scenarios where multiple experts are merged?
•	In Appendix H.3, the authors briefly show that trainable routers perform worse, could the authors explore other routing strategies specifically designed for multi-expert selection?
•	Beyond adjusting the calibration parameter $rho$, what principled approaches could ensure more balanced expert utilization while maintaining performance?
•	How does expert imbalance affect the theoretical guarantees of Theorem 1, especially when some experts receive minimal training?

**Ethical Concerns:**

["NO or VERY MINOR ethics concerns only"]

**Final Justification:**

I have read the rebuttal and also looked at the other reviews. I don't need further clarifications at this point. I am satisfied with the rating I have already provided but increased the scores slightly.

**Limitations:**

yes

**Paper Formatting Concerns:**

no formatting issues

**Quality:**

4

**Strengths And Weaknesses:**

Strengths:
•	The motivation is sound, which is based on straightforward empirical observation. This points out the deficiency of current approaches and poses a natural goal to solve the explicitly existing problem.
•	The paper presents two complementary solutions addressing the same problem from different angles, where KDEM operates in output space using knowledge distillation, while PPEM works in parameter space using theoretical proximity principles.
•	The theoretical validation for PPEM is sound and solid. Theorem 1 provides a solid mathematical basis for PPEM, proving that when MLP expert parameters converge pairwise, the merged MLP output approximates the convex combination of individual expert outputs. This insight directly translates into PPEM that maintains expert diversity while reducing parameter interference.
•	This work demonstrates substantial empirical contributions. Evaluation on 38 datasets spanning diverse domains with rigorous zero-shot evaluation protocols ensures robustness and generalizability. The proposed approach achieves significant improvements (5.27% and 4.35% recall improvement on Link2) while maintaining computational efficiency nearly identical to single-expert inference. As for efficiency, both proposed expert merging strategies introduce minimal computational overhead - KDEM adds only 4.8% training time, while PPEM has negligible increase, making them practically deployable.
Weaknesses:
•	The paper inherits AnyGraph's routing mechanism without questioning its optimality for multi-expert scenarios.
•	The expert imbalance problem (some experts rarely selected) is acknowledged but not systematically addressed.
•	Insufficient analysis of when and why the methods work well across different graph types. A more principled analysis of what graph characteristics, e.g., size, feature dimensionality, domain, make certain datasets more amenable to the multi-expert approach would be good.

---

> ### Author Rebuttal · Authors · 2025-07-27
>
> We sincerely thank the reviewer for this thorough evaluation and positive feedback. We appreciate your recognition of our paper's strengths, particularly the sound motivation, complementary solution approaches, solid theoretical foundation, and substantial empirical validation. Below we address the concerns and questions in detail.
> > *W1: The paper inherits AnyGraph's routing mechanism without questioning its optimality for multi-expert scenarios.*
> >
> > *Q1: Have the authors investigated whether the non-parametric routing mechanism is actually optimal for scenarios where multiple experts are merged? In Appendix H.3, the authors briefly show that trainable routers perform worse, could the authors explore other routing strategies specifically designed for multi-expert selection?*
>
> We thank the reviewer for this insightful concern. Here, we handle W1 and Q1 together, and our responses are as follows:
> - **Rationale for Our Initial Choice:** Our paper's primary contribution is the development of novel expert merging strategies (KDEM and PPEM). To isolate and clearly demonstrate their effectiveness, we deliberately inherited the routing mechanism from our main baseline, AnyGraph. We believe this non-parametric router provides a strong and generalizable foundation; its competence score calculation is independent for each expert, allowing the selection to be naturally extended from a top-1 to a top-$k$ setting. This provides a fair and controlled basis for evaluating our core contributions.
> - **Investigation of Alternative Routers:** However, we agree with the reviewer that investigating alternatives is crucial for a complete analysis. Prompted by this valuable feedback, we have conducted new experiments to supplement our initial investigation of a trainable router (Appendix H.3). We selected another well-established strategy, Expert Choice Routing [1], adapted it for our framework, and evaluated its performance with our PPEM method. The results, averaged on the Link2 group, are as follows:
>
> | Method                        | Recall | NDCG  |
> |-------------------------------|--------|-------|
> | PPEM w/ Expert Choice Routing | 50.06  | 30.91 |
> | PPEM                          | 50.77  | 31.93 |
>
> - **Discussion of Results:** The results show that Expert Choice Routing, while a powerful technique, underperforms compared to the original non-parametric router in our setting. This finding reinforces our hypothesis (lines 870–872) that the heuristic non-parametric router may be particularly effective at assigning structurally similar graphs to the same experts, making it a surprisingly strong baseline for graph-based MoE.
> These new results, combined with our initial findings in Appendix H.3 that a standard trainable router also underperforms, further validate our choice of routing mechanism as a solid foundation for our study. We are grateful for the reviewer's suggestion, as it has allowed us to strengthen the paper by demonstrating the robustness of our findings across different routing paradigms. We will add this new analysis to the appendix.
>
>
> [1]. Yanqi Zhou et al. Mixture-of-Experts with Expert Choice Routing. *NeurIPS 2022.*
>
> > *W2: The expert imbalance problem (some experts rarely selected) is acknowledged but not systematically addressed.*
> >
> > *Q2: Beyond adjusting the calibration parameter $\rho$, what principled approaches could ensure more balanced expert utilization while maintaining performance?*
>
> We thank the reviewer for this valuable question. We respond to W2 and Q2 together because they are fully correlated. Addressing the expert imbalance problem without compromising performance is indeed a critical and challenging research direction in MoE models, especially in the LLM community, and we appreciate the opportunity to elaborate on this.
>
> As the reviewer noted, our investigation in Appendix H.2 showed that naively enforcing balance by adjusting the calibration parameter $\rho$ can lead to suboptimal expert assignments and degrade performance. This highlights a fundamental tension between forcing uniform routing and allowing the model to select the most specialized expert for optimal performance. Our intention in exposing this challenge of MoE-based GFMs is to encourage the graph learning community to actively seek solutions. Drawing on the successful experiences of the LLM community, there are currently several promising approaches worth exploring:
> - Auxiliary Load Balancing Loss: This is the most common strategy, used in seminal works like [2][3]. An auxiliary loss term is added to the main training objective to penalize router imbalance.
> - Auxiliary-Loss-Free Balancing Strategies: More recent work has focused on achieving balance without auxiliary losses, for instance, the work by Wang et al. [4], which we cite, proposes a redesign of the routing algorithm to deterministically balance the load across experts while prioritizing high-confidence assignments.
> - Expert Pruning Techniques: An alternative perspective is to treat chronic underutilization as a signal of expert redundancy.  The method proposed by Lu et al. [5] could be used to identify and remove less useful experts. This would not enforce balance but would improve computational efficiency by removing the "dead weight" of unused parameters.
>
> [2]. Noam Shazeer et al. Outrageously Large Neural Networks: The Sparsely-Gated Mixture-of-Experts Layer. *ICLR 2017.*
>
> [3]. William Fedus et al. Switch Transformers: Scaling to Trillion Parameter Models with Simple and Efficient Sparsity. *JMLR 2022.*
>
> [4]. Lean Wang et al. Auxiliary-loss-free load balancing strategy for mixture-of-experts. *arXiv 2024.*
>
> [5]. Xudong Lu et al. Not All Experts are Equal: Efficient Expert Pruning and Skipping for Mixture-of-Experts Large Language Models. *ACL 2024.*
>
> > *W3: Insufficient analysis of when and why the methods work well across different graph types. A more principled analysis of what graph characteristics, e.g., size, feature dimensionality, domain, make certain datasets more amenable to the multi-expert approach would be good.*
>
> This is an excellent suggestion. We agree that a more principled analysis of what graph characteristics lead to performance gains would strengthen the paper.
> - **Initial Analysis:** We have provided an initial analysis in Section 5.2, where we note that our methods give a particularly strong boost to datasets in the E-commerce domain. Additionally, in Appendix F, we discuss how our multi-expert approach helps mitigate the class imbalance issue in the Arxiv dataset by leveraging diverse expert perspectives.
> - **Action Taken:** To provide a deeper analysis as requested, we correlate the performance gains from Table 5 with the graph properties listed in Table 4. Comparing the results of Arxiv and Arxiv-t (the raw text features are provided and are encoded by a language model to get the node feature matrix), where our methods achieve a performance boost on Arxiv-t against AnyGraph but degrade on Arxiv, we guess that, the feature quality is essential for the benefits of expert merging. Following this insight, we can explain why the E-commerce datasets such as Fitness, Goodreads and Photo, benefit more from expert merging since they all use the semantic-rich node features. This may be due to the high-quality features leading to more accurate expert assignment and making it easier to merge features. We will add this analysis to Appendix F.
>
> > *Q3: How does expert imbalance affect the theoretical guarantees of Theorem 1, especially when some experts receive minimal training?*
>
> We appreciate the reviewer's careful consideration of this theoretical question.
> The premise of Theorem 1 is conditional; it applies specifically to the set of experts selected for merging at a given time. The theorem requires that the parameters of the selected top-$k$ experts are close to each other. It does not impose any requirements on the state of the unselected experts.
>
> Our method, PPEM, works in synergy with the upstream router in a way that self-selects for the theorem's conditions:
>
> The router's primary function is to identify the most competent experts for a given graph. As our analysis in Appendix H shows, the router consistently assigns high competence scores to a small subset of specialized experts. Chronically under-trained experts receive low scores and are therefore very rarely included in the selected top-$k$ set for each graph.
> Consequently, the EMA update in PPEM (Eq. 13) is almost always applied to a set of well-trained, task-relevant experts, making them amenable to the convergence condition required by Theorem 1.

---

> > ### Comment · Reviewer_Etth · 2025-08-08
> >
> > Thanks to the authors for your work conducting extra experiments to address my comments. I think this is a nice paper, and it should be accepted.

---

### Official Review · Reviewer_WjEh · 2025-07-02

**Clarity:** 3
**Significance:** 3
**Originality:** 4
**Rating:** 5
**Confidence:** 4

**Summary:**

Based on the current Graph MoE models like Anygraph, this works provide detailed discussitions on Expert Selection and Merging. After analyzation on this, this work propose two improved expert
merging strategies, KDEM and PPEM. Further Experiments show that they can achieve better performance than naive experts merging mechanism.

**Questions:**

1. L183 : "Since the performance of the top-k experts’ ensemble is excellent, we treat it as a teacher model to teach the parameter-fused expert." How is the "top-k" determined in a MoE model?

2. KD-based methods for merging graph-based models have been widely discussed, such as in ParetoGNN[1](during training) or WAS[2](during fine-tuning). Could you please provide more details about your design? Why your approach is unique or different?

---

[1]ICLR'23 https://arxiv.org/abs/2210.02016

[2]ICLR'24 https://arxiv.org/abs/2403.01400

**Ethical Concerns:**

["NO or VERY MINOR ethics concerns only"]

**Final Justification:**

Great work. Good Paper. So I will keep my score.

**Limitations:**

N.A.

**Paper Formatting Concerns:**

N.A.

**Quality:**

4

**Strengths And Weaknesses:**

Strengths:

1. The authors address an important and previously under-explored problem — the selection and merging of experts in Graph MoE models

2. Preliminary analysis is both reasonable and insightful.
The method proposed by the authors is effective and powerful.

Weaknesses:

Will these processes (KDEM or PPEM) be harmful to the training time? It seems quite difficult to converge, especially when KD is introduced in the middle of the training process.

---

> ### Author Rebuttal · Authors · 2025-07-29
>
> We are grateful to the reviewer for this positive evaluation. We are encouraged that the reviewer found the problem we addressed to be important (S1), our analysis insightful (S2), and our proposed methods effective (S2). We address the specific weakness and questions below.
>
> > *Weakness: Will these processes (KDEM or PPEM) be harmful to the training time? It seems quite difficult to converge, especially when KD is introduced in the middle of the training process.*
>
> We understand the reviewer's concern regarding training time and convergence. We are pleased to clarify that both KDEM and PPEM were designed for efficiency and converge effectively.
>
> - **On Training Time:** Our computational efficiency analysis in Section 5.4 shows that the additional overhead is minimal. As stated, "KDEM incurs a marginal increase in training time over standard expert merging (averaging 4.8%), whereas PPEM exhibits almost no increase". This is visually demonstrated in Figure 4, where the training times for our methods are very close to the Top-1 and standard merging baselines, and significantly faster than expert ensembling.
>
> - **On Convergence:** The strong and consistent performance improvements reported in our main results (Tables 1 and 2) serve as direct evidence that our methods converge successfully to a high-performing state. Furthermore, applying knowledge distillation periodically (rather than at every step) in KDEM actually promotes stable convergence. This is because the student model is not forced to chase a constantly changing teacher target, allowing for more consistent gradient updates, a principle we highlight in our discussion of KDEM's design.
>
> > *Q1: L183 : "Since the performance of the top-$k$ experts’ ensemble is excellent, we treat it as a teacher model to teach the parameter-fused expert." How is the "top-$k$" determined in a MoE model?*
>
> Thank you for the question. In our framework, $k$—the number of experts to select for merging—is a hyperparameter. The process is as follows:
>
> For a given input graph, the routing mechanism calculates a competence score for each of the $K$ experts in the pool. We then simply select the $k$ experts that have the highest competence scores.
>
> We provide an empirical analysis of this hyperparameter in Figure 10, which shows that a small value of $k$ (e.g., 2 or 3) is a robust choice that consistently yields strong performance. This provides a practical guideline for its selection.
>
> > *Q2: KD-based methods for merging graph-based models have been widely discussed, such as in ParetoGNN[1](during training) or WAS[2] (during fine-tuning). Could you please provide more details about your design? Why your approach is unique or different?*
>
> Thank you for your insightful feedback and for highlighting these important related works. We appreciate the opportunity to provide more details about our design and clarify the unique contributions of our proposed method, KDEM.
>
> While all three works involve integrating knowledge from multiple sources, they operate on different principles and architectural levels.
>
> We respectfully clarify that ParetoGNN's methodology is distinct from knowledge distillation. ParetoGNN's core contribution is a multi-task learning framework that reconciles conflicting signals from different self-supervised pretext tasks. It operates in the **gradient space**, using a Multiple Gradient Descent Algorithm (MGDA) to find a Pareto-optimal descent direction that balances the objectives of all tasks simultaneously. In contrast, our KDEM operates in the **output space**. The "teacher" is the soft prediction distribution from an expert ensemble, and the "student" (the merged expert) is trained to mimic this distribution. Therefore, ParetoGNN focuses on balancing training objectives at the gradient level, whereas our work focuses on compressing the predictive knowledge of an expert ensemble into a single, efficient expert.
>
> While both our work and WAS employ knowledge distillation (KD) during a fine-tuning or unified training phase, we address fundamentally different problems with distinct goals and methodologies. Our work focuses on **efficient expert merging within a single Mixture-of-Experts (MoE) model**, whereas WAS focuses on **integrating and selecting from a pool of separate, independently pre-trained models.**
>
> Our contributions are unique in two key ways:
>
> **1. Different Problem & Architecture:** We optimize a single MoE-based GFM. Our goal is to merge the parameters of the top-$k$ experts to efficiently approximate the performance of a computationally expensive expert ensemble. WAS integrates multiple, separate pre-trained models by combining their output distributions during a fine-tuning stage.
>
> **2. Novel KD Application (in KDEM):** The teacher-student dynamic is fundamentally different.
>
> - In our KDEM, the "teacher" is the on-the-fly ensemble of top-$k$ experts, and the "student" is the parameter-merged expert. KD's role is to align the behavior of the efficient merged expert with its powerful ensemble counterpart within a unified training loop.
>
> - In WAS, the "teacher" is a weighted combination of outputs from selected pre-trained models, and the "student" is a separate model being fine-tuned.
>
> In summary, our work introduces new techniques for efficient expert utilization inside an MoE model, while WAS tackles the integration of external models. Our focus on MoE mechanics is a distinct and complementary contribution.

---

> > ### Comment · Reviewer_WjEh · 2025-08-05
> >
> > Thanks for your response. Great work : ) I will keep my score.

---

### Official Review · Reviewer_bgSg · 2025-07-05

**Clarity:** 4
**Significance:** 3
**Originality:** 3
**Rating:** 5
**Confidence:** 4

**Summary:**

Authors propose two enhanced expert merging strategies for Graph Foundation Models (GFMs) to improve performance while maintaining computational efficiency. The authors aim to address the challenges of the two extreme versions -- (1) a single top-ranked expert underutilizing knowledge from multiple experts and (2) full ensemble strategies with huge computational overhead.

In particular, the authors perform an in-depth analysis of an MoE-based GFM and discover that the second- and third-ranked experts often outperform the top-ranked expert selected by the router, revealing valuable but untapped expertise. Motivated by this insight, authors propose two methods: (i) Knowledge Distillation Enhanced Expert Merging (KDEM), which aligns a fused expert’s behavior with that of an expert ensemble via periodic distillation, and (ii) Parameter Proximity Enhanced Expert Merging (PPEM), which reduces parameter distances among top-k experts so that merging approximates ensemble outputs while preserving diversity. These approaches are designed to harness the strengths of multiple experts efficiently.

Authors perform extensive experiments on 38 graph datasets, evaluating their methods under zero-shot and few-shot settings against baseline GFMs, classical GNNs, and standard expert merging. Results show that KDEM and PPEM consistently improve performance over single-expert routing and standard merging while maintaining low computational costs. They further provide ablation studies, theoretical analysis, and routing visualizations to validate the effectiveness and efficiency of the proposed methods.

**Questions:**

- What exactly does  `-EMA` in Table 3 indicate?
- It would be great if Figure 4 can be readable along with performance as well (i.e. clear inteprataion over time / top K / performance)
- It would be great to see how different experts are trained between KDEM and PPEM, given that PPEM adds the restriction of not diverging too much.

**Ethical Concerns:**

["NO or VERY MINOR ethics concerns only"]

**Limitations:**

Yes

**Paper Formatting Concerns:**

No concern

**Quality:**

4

**Strengths And Weaknesses:**

* Strengths
- Authors carefully analyze the behavior of existing MoE-based GFMs and identify the important challenge that second- and third-ranked experts often outperform the top expert chosen by routing. This empirical observation is clearly presented and motivates the necessity for improved multi-expert utilization, which grounds the rest of the work in a real and measurable problem.
- Authors provide a clean and rigorous rationale of the proposed solutions, combining empirical findings and theoretical insights—for example, proving that parameter proximity enables parameter merging to approximate expert ensembles. This thoroughness makes the proposed approaches well justified and not merely heuristic.
- The introduction of knowledge distillation into expert merging (KDEM) is novel to tackle the challenges in the ensemble learning of GFMs. The method intuitively bridges the gap between the efficiency of expert merging and the performance of expert ensembles, and the authors provide practical implementation details that make it feasible.
- The authors perform experiments on 38 diverse graph datasets across domains, under both zero-shot and few-shot settings. This broad scope of evaluation, along with detailed ablation studies, strongly supports the validity and robustness of the proposed methods.
- The paper is well-written and organized, making a technically complex topic accessible. Figures, tables, and theoretical explanations are presented in a way that is easy to follow, which enhances the reader’s understanding.


Weakness
- The core contributions of the paper lie in improving expert merging strategies within an existing MoE-based GFM framework rather than proposing a fundamentally new architecture or a completely novel GFM paradigm. While the enhancements are well-justified and practically useful, some readers might view the work as an incremental improvement rather than a groundbreaking innovation.
- Despite recognizing the limitations and risks of parameter averaging in expert merging (e.g., parameter interference due to expert specialization), the authors focus solely on improving parameter averaging rather than exploring fundamentally different fusion mechanisms.
- The PPEM method intentionally brings expert parameters closer together to enable more effective merging, while the main purpose of MOE is to cover multi-modal diversities. While proper regularization might work well in practice to balance between these two conflicting goals, the authors need to address this concern.

---

> ### Author Rebuttal · Authors · 2025-07-28
>
> We sincerely thank the reviewer for this detailed and positive assessment of our work. We are delighted that the reviewer found our paper to be well-motivated (S1), technically solid and novel (S2、S3), comprehensively and robustly evaluated (S4), and clearly written (S5). We appreciate the insightful questions and address each point below.
>
> > *W1: The core contributions of the paper lie in improving expert merging strategies within an existing MoE-based GFM framework rather than proposing a fundamentally new architecture or a completely novel GFM paradigm. While the enhancements are well-justified and practically useful, some readers might view the work as an incremental improvement rather than a groundbreaking innovation.*
>
> We thank the reviewer for this thoughtful perspective on the positioning of our work. We agree that our work focuses on a specific, critical component of MoE-based models rather than proposing a new GFM architecture from scratch. We respectfully clarify that this targeted focus is a significant strength and a crucial contribution to the field.
>
> MoE is becoming a cornerstone for large-scale models across many domains, such as LLMs (DeepSeek, Qwen3), vision models (V-MoE), and multi-modal models (Gemini 2.5 Pro). While models like AnyGraph have pioneered the use of MoE in the graph space, the fundamental question of how to efficiently utilize the collective knowledge of multiple experts remains a key bottleneck. Our paper tackles this precise challenge head-on. We are, to our knowledge, the first to move beyond simple averaging and propose enhanced expert merging strategies specifically for this context. Our methods provide principled, effective, and efficient solutions to this critical bottleneck.
>
> Therefore, **while our work builds upon an existing framework, it solves a fundamental problem that will be faced by any future MoE-based GFM.** By providing a blueprint for efficient multi-expert utilization, our research offers a foundational piece of the puzzle that will enable the development of more powerful and scalable graph foundation models. We believe this contribution is both practical and of significant long-term value to the community.
>
>
> > *W2: Despite recognizing the limitations and risks of parameter averaging in expert merging (e.g., parameter interference due to expert specialization), the authors focus solely on improving parameter averaging rather than exploring fundamentally different fusion mechanisms.*
>
> We thank the reviewer for raising an interesting point about exploring different fusion mechanisms. We chose to focus on enhancing parameter averaging because it is the cornerstone of efficient expert merging in MoE models. While other fusion mechanisms exist [1], they often come with increased computational complexity, defeating the primary purpose of merging.
>
> Our contribution lies in making the most computationally efficient paradigm, simple parameter averaging, perform on par with expensive expert ensembling. By demonstrating that this simple mechanism can be elevated to match the performance of complex ensembling through our proposed methods, we provide a practical and powerful toolkit for a common scenario.
>
> [1]. Prateek Yadav et al. TIES-Merging: Resolving Interference When Merging Models. *NeurIPS 2023*.
>
> > *W3: The PPEM method intentionally brings expert parameters closer together to enable more effective merging, while the main purpose of MOE is to cover multi-modal diversities. While proper regularization might work well in practice to balance between these two conflicting goals, the authors need to address this concern.*
>
> We thank the reviewer for raising this critical point. We were fully aware of this potential conflict, which is why we designed PPEM to strike a careful balance between expert similarity and diversity. As detailed in Appendix G.3, our approach has two key features:
>
> 1. Selective Similarity: The parameter-proximity constraint is applied only to the top-$k$ experts selected for a given input. The parameters of the remaining experts are left unchanged, preserving the overall diversity of the expert pool.
>
> 2. Verified Diversity: Our t-SNE visualization in Figure 9 empirically confirms that even after training with PPEM, the full set of 8 experts remains well-separated and specialized in the embedding space, demonstrating that global diversity is maintained.
>
> Therefore, PPEM successfully encourages local similarity for effective merging while preserving the global diversity essential for the MoE architecture.
>
> > *Q1: What exactly does `-EMA` in Table 3 indicate?*
>
> In Table 3, PPEM-EMA refers to an ablated version of our PPEM model where the core Exponential Moving Average (EMA) update mechanism (described in Eq. 13) has been removed. This variant is equivalent to the standard expert merging baseline, where the parameters of the top-$k$ selected experts are simply averaged without any of our proposed enhancements. The performance drop from PPEM to PPEM-EMA thus directly demonstrates the effectiveness of our parameter proximity strategy. We are sorry for any confusion and will clarify this in the final version.
>
> > *Q2: It would be great if Figure 4 can be readable along with performance as well (i.e. clear inteprataion over time / top K / performance)*
>
> This is a great suggestion. For the sake of clarity, we separated the analysis of computational efficiency (Figure 4) from the parameter sensitivity analysis that shows performance vs. $k$ (Figure 10). To make the demonstration clearer, we will add a new plot to the appendix in the final version that directly visualizes performance against training/inference time for different values of $k$.
>
> > *Q3: It would be great to see how different experts are trained between KDEM and PPEM, given that PPEM adds the restriction of not diverging too much.*
>
> Thank you for this insightful feedback. We will generate a t-SNE visualization of node embeddings from all 8 experts, similar to the existing Figure 9, but this time for a model trained with KDEM. We expect to see that the expert clusters for PPEM are tighter and show less separation than for KDEM.

---

### Official Review · Reviewer_aTL7 · 2025-07-07

**Clarity:** 4
**Significance:** 3
**Originality:** 3
**Rating:** 4
**Confidence:** 4

**Summary:**

This paper addresses a key limitation of the current mixture-of-experts (MoE)-based graph foundation model, which relies solely on the top-ranked expert, by showing that lower-ranked experts (e.g., second- and third-ranked) can often perform better. To more effectively harness the collective knowledge of multiple experts, this paper proposes two novel expert merging techniques: (1) KDEM, which distills knowledge from an ensemble of top-k experts into a parameter-merged expert, and (2) PPEM, which encourages parameter alignment among the selected experts. The resulting model achieves strong performance on several downstream tasks in a near-consistent manner and improves training and inference efficiency over naive expert ensembling.

**Questions:**

Please refer to the main review for detailed comments and suggestions.

One additional question: The paper applies KDEM and PPEM only individually. Was combining the two techniques considered? If so, why was this not pursued? If it was not considered, do the authors believe the two techniques could be complementary if used together?

**Ethical Concerns:**

["NO or VERY MINOR ethics concerns only"]

**Final Justification:**

The author response addressed my concerns about the comparison with existing expert merging strategies and the choice of total and selected number of experts. In light of these improvements, I have increased my rating.

**Limitations:**

yes

**Quality:**

3

**Strengths And Weaknesses:**

[Strengths]

S1. The paper is well-motivated and easy to follow. It builds on an interesting observation: in mixture-of-experts (MoE) models, the second- and third-ranked experts often outperform the top-ranked one. This highlights (1) a limitation of the recent MoE-based graph foundation model, AnyGraph, which uses only the top-ranked (i.e., most competent) expert, and (2) the need to incorporate the knowledge embedded in multiple experts.

S2. The paper introduces two interesting and novel expert merging techniques: Knowledge Distillation Enhanced Expert Merging (KDEM) and Parameter Proximity Enhanced Expert Merging (PPEM). KDEM leverages knowledge distillation, using an ensemble of the top-k experts as the teacher and the parameter-merged expert as the student. PPEM encourages the parameters of the top-k experts selected by the router to move gradually in the same direction.

S3. The proposed method near-consistently outperforms several existing graph foundation models and graph learning methods on downstream tasks. Ablation studies demonstrate the effectiveness of the expert merging techniques, as well as internal components such as the use of EMA. Compared to naive expert ensembling, the method also achieves improved training and inference efficiency, as supported by experimental results.


[Weaknesses]

W1. The paper currently lacks a comparison between the proposed method and existing expert merging strategies. The experiments focus on comparing the proposed method against graph foundation models and graph learning methods for graph tasks such as link prediction and node classification. However, the core technical contributions of this work lie in the development of expert merging techniques, which are not specific to graph data or graph learning methods. Given the existence of prior expert merging approaches such as SMEAR [28] and Lory [51], a comparison with those methods is necessary to establish the proposed technique as a new state of the art.

W2. The number of total and selected experts (denoted as K and k, respectively) has a significant impact on performance, as demonstrated in the experiments. However, these values must be manually specified prior to training, which may be non-trivial in the context of graph foundation models spanning diverse datasets and tasks. It would strengthen the paper to provide guidance or at least discussion on how to choose these values effectively and efficiently.

W3. The generalizability of the proposed method to new domains and tasks appears limited or remains unverified. A core requirement for a graph foundation model (GFM) is the ability to generalize to unseen domains and tasks. While the proposed GFM builds on AnyGraph and improves it by leveraging multiple experts, it remains unclear whether the selected experts can perform well under significant domain or task shifts. The current experiments use two dataset groups (Link 1 and Link 2), but these are relatively homogeneous in both domain and task. It would strengthen the paper to include evaluations on more diverse domains and tasks to better assess the model’s robustness in generalization scenarios, which are central to the purpose of GFMs.

---

> ### Author Rebuttal · Authors · 2025-07-28
>
> We thank the reviewer for this detailed and constructive feedback. We are encouraged that the reviewer found the paper well-motivated and easy to follow (S1), the proposed techniques (KDEM and PPEM) to be interesting and novel (S2), and the empirical results and efficiency gains to be strong (S3). We appreciate the critical suggestions for improvement, which we address below.
> > *W1: The paper currently lacks a comparison between the proposed method and existing expert merging strategies ... Given the existence of prior expert merging approaches such as SMEAR [28] and Lory [51], a comparison with those methods is necessary to establish the proposed technique as a new state of the art.*
>
> We thank the reviewer for raising this important point and for suggesting these relevant baselines. We would like to clarify how our work is positioned relative to these methods.
>
> The approaches of SMEAR and Lory are indeed foundational to recent work in expert merging. Both propose a soft merging of experts where the parameters of all experts are fused via a weighted average based on routing weights. Their primary motivation is to create a fully differentiable training process that avoids the challenges of discrete router optimization. We note that **the expert merging strategy of Lory is the same as SMEAR**, with the difference of application scenario (encoder-decoder v.s. decoder-only architecture).
>
> Our "vanilla expert merging" baseline, shown in Figure 8 and Table 5, is inspired by this principle but adapted to the top-$k$ routing framework. Specifically, it performs a parameter merge of the top-$k$ experts selected by the router's gating mechanism. Therefore, the expert merging strategy used by SMEAR and Lory (i.e., $k=K$) is a special case of our adapted vanilla expert merging approach.
>
> To provide a more direct and comprehensive comparison as requested, we report the top-8 (in our case, $K$=8) expert merging results following the SMEAR/Lory methodology on the whole Link2 group datasets:
>
> | Method             | Recall | NDCG  |
> |-------------------|--------|-------|
> | SMEAR/Lory-style Merge | 48.52  | 29.46 |
> | KDEM              | 51.69  | 32.60 |
> | PPEM              | 50.77  | 31.93 |
>
> As the results clearly show, our proposed KDEM and PPEM methods significantly outperform the SMEAR/Lory-style baseline. We hypothesize this is because fusing all experts—including those with very low competence for a given graph—can introduce noise and parameter interference from irrelevant specialists. Our approach is more targeted: it first identifies the most relevant experts via top-$k$ routing and then applies our enhanced merging strategies to effectively combine their specialized knowledge. This focused enhancement proves to be more effective than a simple fusion of all available experts.
>
> > *W2: The number of total and selected experts (denoted as $K$ and $k$, respectively) has a significant impact on performance ... It would strengthen the paper to provide guidance or at least discussion on how to choose these values effectively and efficiently.*
>
> We thank the reviewer for this insightful suggestion. Providing clear guidance on selecting these key hyperparameters is crucial for practitioners, and we will add a dedicated section in the appendix to address this based on our findings.
>
> Our recommendations are as follows:
>
> 1. Guidance on Choosing the Total Number of Experts ($K$)
>
> The choice of $K$ involves a trade-off between model capacity and efficiency. A larger $K$ allows the model to learn a more diverse set of specializations, which is beneficial when training on a wider range of domains. However, it also increases the model's parameter count and can exacerbate the expert imbalance issue. A smaller $K$ is more parameter-efficient but may not fully capture the data's heterogeneity.
>
> **Recommendation:** Based on our experiments across 38 diverse datasets (Table 10), we find that $K=8$ provides a good balance, effectively covering multiple domains without being excessively large.
>
> 2. Guidance on Choosing the Number of Selected Experts ($k$)
>
> We propose two strategies for selecting $k$, catering to different experimental constraints:
>
> **Empirical Guideline (For Optimal Performance):** We recommend a preliminary analysis based on the performance of individual experts, similar to our study in Figure 1. After an initial training run, one can evaluate the performance of each expert ranked by competence score. The value of $k$ should be chosen to include the top-performing experts while excluding those that cause a sharp performance drop. For instance, in Figure 1, performance drops after the 5th expert, suggesting an optimal $k\leq5$. This avoids merging poorly performing experts whose parameters could introduce noise and interference.
>
> **Efficient Heuristic (As a Default):** In scenarios where extensive tuning is not feasible, our results in Figure 10 (the first column) show that using a small value like $k=2$ or $k=3$ serves as a robust and effective default. This consistently provides a significant performance lift over using a single expert ($k=1$) without incurring the high computational cost or risk of noise associated with larger values of $k$.
>
> > *W3: The generalizability of the proposed method to new domains and tasks appears limited or remains unverified ... it remains unclear whether the selected experts can perform well under significant domain or task shifts. The current experiments use two dataset groups (Link 1 and Link 2), but these are relatively homogeneous in both domain and task. It would strengthen the paper to include evaluations on more diverse domains and tasks ...*
>
> We thank the reviewer for this crucial comment. We completely agree that rigorous evaluation of generalizability across diverse domains and tasks is paramount for advancing GFMs.
>
> **1. On the Diversity of Domains in Our Current Evaluation:**
>
> We would like to respectfully highlight the significant domain diversity already present in our experimental setup. Our evaluation, which involves training on one group of datasets and zero-shot testing on another, was designed to explicitly test this cross-domain generalization. For example:
>
> - Domains unique to Link1 group: Email Networks (Email-Enron), Trust Networks (Soc-Epinions1), and Internet peer-to-peer network (P2P-Gnutella06).
> - Domains unique to Link2 group: Web Graphs (Web-Stanford) and Road Networks (RoadNet-PA).
>
> Our model's strong performance when transferred between these groups demonstrates a robust ability to generalize to entirely unseen graph domains. The inclusion of common domains like "Academic" and "E-commerce" in both groups is a deliberate choice since they are widespread, analogous to how LLMs are pre-trained on broad web data that contains domains also seen in downstream evaluations (e.g., math, code). This ensures the model learns both domain-specific and widely generalizable patterns.
>
> **2. On the Diversity of Tasks:**
>
> Regarding task diversity, our work focuses on node-level (classification) and edge-level (link prediction) tasks. This aligns with the evaluation protocol of our primary baseline, AnyGraph, and much of the current GFM literature. We agree that a truly universal GFM should also excel at graph-level tasks (e.g., graph classification, regression). We consider the extension and evaluation of our methods on these tasks an important direction for future work, which we have noted in our limitations section (Appendix L).
>
> **3. On the State of GFM Benchmarking:**
>
> We also acknowledge the reviewer's sentiment about the need for even broader evaluations. A significant challenge in the current GFM landscape is the lack of a standardized, large-scale benchmark that spans a vast number of disparate domains and tasks. In fact, our collection of 38 datasets is one of the more comprehensive evaluation suites used in GFM research to date. We see our work not only as a methodological contribution but also as a step towards highlighting the need for more extensive, community-driven GFM benchmarks, and we believe our findings provide a strong baseline for such future evaluations.
>
> > *Question: The paper applies KDEM and PPEM only individually. Was combining the two techniques considered? If so, why was this not pursued? If it was not considered, do the authors believe the two techniques could be complementary if used together?*
>
> We thank the reviewer for this forward-looking idea. The potential for combining KDEM and PPEM is indeed an exciting prospect.
>
> In the current work, we presented KDEM and PPEM individually to clearly isolate and analyze their distinct mechanisms. **As discussed in Appendix I**, the two methods achieve the same goal—elevating the performance of a merged expert to that of an expert ensemble—through different means: KDEM operates in the output space while PPEM operates in the parameter space. This indicates that they may possess complementary advantages and potentially offers synergistic benefits if combined. A potential hybrid approach could work as follows:
>
> - **PPEM as a Regularizer:** The PPEM method, by applying an EMA to the selected experts' parameters, can be seen as a strong regularizer that encourages the "student" model (the merged expert) to be inherently more similar to the "teacher" model (the expert ensemble) even before distillation occurs.
>
> - **KDEM as a Fine-Tuning Mechanism:** With PPEM creating a more stable and better-aligned student, the knowledge distillation from KDEM could then act as a more focused fine-tuning step. The distillation loss might be smaller and converge faster, as PPEM has already done much of the work to close the gap between the student and teacher.
>
> Given the scope of this work, we chose to present them as two distinct and effective strategies. However, we consider the investigation of a hybrid KDEM-PPEM model to be a promising direction for future research.

---

> > ### Comment · Reviewer_aTL7 · 2025-08-04
> >
> > Thank you for your response. I have a follow-up question regarding the datasets.
> >
> > In the author response, the collection of 38 datasets appears to be presented as a contribution of this work, in the form of a benchmark. However, the datasets seem to be the same as those used in the previous AnyGraph paper. Could the authors clarify whether there are any new contributions or changes to the dataset collection compared to what was proposed in AnyGraph?

---

> > > ### Author Response · Authors · 2025-08-04
> > >
> > > We thank the reviewer for this follow-up question and appreciate the opportunity to provide a clear and direct clarification.
> > >
> > > To be perfectly clear: we are not claiming the collection of 38 datasets as a contribution of our work. The reviewer is absolutely correct that we adopt the same comprehensive dataset suite introduced and used by the authors of AnyGraph. We apologize if our previous wording created any ambiguity on this point.
> > >
> > > Our intention in stating that "*our collection of 38 datasets is one of the more comprehensive evaluation suites*" was not to claim credit for its creation. Rather, our goal was to address the reviewer's original, valid concern about generalizability. By choosing to evaluate our methods on this existing, large-scale benchmark, we aimed to demonstrate that our proposed enhancements are tested rigorously against the current state-of-the-art standards for GFM evaluation.

---

### Decision · Program_Chairs · 2025-09-17

**Decision:**

Accept (poster)

**Comment:**

This manuscript considers the problem of mixture-of-experts in building graph foundation models. The authors begin with an intriguing empirical finding: that using the top-ranked expert for each graph does not necessarily give the best performance. The authors proceed to develop two nuanced expert merging strategies that appear to provide improved performance in a series of empirical studies.

This work was received with significant enthusiasm from the reviewers, who found the problem interesting, the proposed methods well-motivated and technically sound, and the experiments convincing. I would go further to say that the reviewers expressed the feeling of having come away after reading the paper with new found _insight_ that they did not have before.

Although the reviewers expressed some minor concerns during the review period, these were adequately addressed during the discussion period.

In short, this appears to be high-quality work that should be presented at the conference without question.